# The neutrotime transcriptional signature defines a single continuum of neutrophils across biological compartments

Ricardo Grieshaber-Bouyer[1,2], Felix A. Radtke [1,2], Pierre Cunin[1], Giuseppina Stifano[1], Anaïs Levescot[1], Brinda Vijaykumar[3], Nathan Nelson-Maney [1], Rachel B. Blaustein[1], Paul A. Monach[1,4,21], Peter A. Nigrovic [1,5,21 ✉] & ImmGen Consortium*

Neutrophils are implicated in multiple homeostatic and pathological processes, but whether functional diversity requires discrete neutrophil subsets is not known. Here, we apply single-cell RNA sequencing to neutrophils from normal and inflamed mouse tissues. Whereas conventional clustering yields multiple alternative organizational structures, diffusion mapping plus RNA velocity discloses a single developmental spectrum, ordered chronologically. Termed here neutrotime, this spectrum extends from immature pre-neutrophils, largely in bone marrow, to mature neutrophils predominantly in blood and spleen. The sharpest increments in neutrotime occur during the transitions from pre-neutrophils to immature neutrophils and from mature marrow neutrophils to those in blood. Human neutrophils exhibit a similar transcriptomic pattern. Neutrophils migrating into inflamed mouse lung, peritoneum and joint maintain the core mature neutrotime signature together with new transcriptional activity that varies with site and stimulus. Together, these data identify a single developmental spectrum as the dominant organizational theme of neutrophil heterogeneity.

[1] Division of Rheumatology, Inflammation, and Immunity, Brigham and Women's Hospital, Harvard Medical School, Boston, MA, USA. [2] Department of Medicine V, Hematology, Oncology and Rheumatology, Heidelberg University Hospital, Heidelberg, Germany. [3] Division of Immunology, Department of Microbiology and Immunobiology, Harvard Medical School, Boston, MA, USA. [4] Rheumatology Section, VA Boston Healthcare System, Boston, MA, USA. [5] Division of Immunology, Boston Children's Hospital, Harvard Medical School, Boston, MA, USA. [21]These authors jointly supervised this work: Paul A. Monach, Peter A. Nigrovic. *A list of authors and their affiliations appears at the end of the paper. ✉email: peter.nigrovic@childrens.harvard.edu

Neutrophils are important innate immune effector cells that participate in biological processes including immune defense, tissue repair, cancer, coronary artery disease, and autoimmunity[1,2]. Engagement in such a broad array of pathophysiological conditions suggests corresponding phenotypic heterogeneity. Indeed, neutrophils vary greatly with respect to nuclear morphology, granule contents, buoyancy, surface markers, and migratory, phagocytic, and suppressor functions[3]. Yet in comparison with other immune cells, subsets of neutrophils remain poorly established, at least in part because it has proven difficult to distinguish discrete subtypes of neutrophils against the complex background of differential maturation, priming and activation. It is therefore unknown whether neutrophils are best conceptualized as a mélange of subsets or as a phenotypic continuum.

The organizational structure of the neutrophil population has practical implications. Neutrophils contribute to multiple immune-driven diseases, as reflected in the efficacy of neutrophil depletion or blockade in experimental models[4–6]. However, the evolutionary conservation of neutrophils reflects their pivotal contribution to immune defense and other homeostatic functions, rendering broad-based neutralization therapeutically unappealing[3,7–11]. Neutrophil heterogeneity could provide opportunities to target pathogenic neutrophil activity, while leaving beneficial functions relatively undisturbed[12]. It is thus important to understand the ontological relationships among neutrophils with divergent phenotypes. In particular, it is essential to establish whether the neutrophil population is characterized by discrete developmental branches, driven for example by distinct transcription factors, or instead reflects a single main sequence of neutrophil maturation from which individual cells diverge as they encounter environmental exposures. Single-cell technologies have rapidly emerged as a powerful platform to examine the transcriptional landscape of thousands of cells simultaneously, offering an unbiased way to study the relationships within a population[13]. While a low amount of transcriptome renders neutrophils challenging to study using this methodology, data are emerging with respect to developing neutrophils and neutrophils from cancerous tissues[14–17]. Availability of high-resolution transcriptomic data from neutrophils from healthy donors and inflamed tissues remains limited.

Here, as part of the Immunological Genome (ImmGen) Project, we employed droplet-based single-cell RNA-seq (scRNA-seq) to profile more than 17,000 mouse neutrophils from bone marrow, blood and spleen as well as from blood, joint, lung and peritoneum from animals undergoing experimental sterile inflammation. We demonstrate that different neutrophil states in healthy mice can be projected onto a single continuum, termed here neutrotime, characterized by clearly defined poles separated by a smooth transcriptomic shift. Using RNA velocity, Monocle analysis and transcription factor mapping, we show that this main sequence has no major branches. Human neutrophils exhibit a concordant transcriptomic pattern. Neutrophils from inflamed mouse tissues deviate from neutrotime in a manner that varies with site and stimulus. These findings reveal a single developmental continuum as the dominant organizational theme underlying neutrophil heterogeneity.

## Results

**Single-cell RNA-seq in neutrophils is technically feasible**. We sorted Ly6G-positive CD11b-positive neutrophils from bone marrow, peripheral blood, and spleen of healthy 6–8-week-old male B6 mice in two independent experiments (Fig. 1a). The sort strategy for all tissues is shown in Supplementary Fig. 1. ScRNA-seq was performed using the 10X platform. Hashtag oligomers

were used to multiplex cells across inflammatory conditions (Supplementary Fig. 2a–d). To eliminate contaminating cells, we applied a multinomial model to the raw unique molecular identifier (UMI) counts to assign cells to one of 249 ImmGen reference populations, excluding cells that belonged to other recognizable lineages[18]. Although the lower RNA content of neutrophils compared to other leukocytes complicates scRNA-seq[19], after rigorous quality control (excluding cells expressing >5% mitochondrial transcripts, doublets with number of transcripts >99%ile, number of transcripts <1%ile), we could retain 4985 neutrophils from blood, 4504 from bone marrow and 3183 from spleen (12,672 total healthy cells) with a median of 1455 unique molecular identifiers (UMIs) and 515 genes per cell (Supplementary Fig. 2e–h). Concordance between datasets enabled pooled analysis (Supplementary Fig. 2i–j). Excluding transcripts expressed in fewer than five cells, we retained 10,900 robustly expressed transcripts, filtered down to 3322 transcripts for data integration and downstream analysis using a residual variance cutoff >1.3.

**Neutrophil cluster abundance varies across tissues**. To reduce the dimensionality of this dataset, we employed Uniform Manifold Approximation and Projection (UMAP) clustering based on the first 20 principal components, using all cells of healthy blood, bone marrow and spleen. Varying the assumptions of k nearest neighbors and resolution parameters of the community detection algorithm within reasonable parameters ($k = 20$; 100; 500 and resolution $= 0.3$; 0.8; 1.5) yielded between 3 and 23 clusters (Supplementary Fig. 3a). Using the finest clustering, and inspecting gene expression associated with each cluster, we eliminated 0.42% of cells from further consideration as debris because of their concentrated ribosomal and mitochondrial transcripts (e.g., *Rps19*, *Rps28*, *mt-Atp6*, *Rps18*, *Rpl32*, *Rpl13*, *mt-Co3*, and *Rpl18a*)[20,21] (Supplementary Fig. 3b). For population-level analysis, we selected arbitrarily the 4-cluster model as a convenient reference (Fig. 1b). Clusters exhibited a recognizable correlation with neutrophil development, beginning with high expression of granule genes such as *Chil3*, *Camp*, *Lcn2*, *Ltf*, and *Mmp9* in population (P) 1 and progressing in P4 to genes associated with mature neutrophils, such as *Csf3r*, encoding the G-CSF receptor, *Il1b*, encoding interleukin-1β, and the chemokine-encoding gene *Ccl6* (Fig. 1c)[22].

We analyzed the frequency of these clusters across tissues. For this analysis, we again chose the same 4 populations, although similar tendencies were identified across models (Supplementary Fig. 3a, b). We found that P1–P3 were most abundant in the bone marrow and P4 most abundant in the blood and spleen, consistent with the expected maturity difference (Fig. 1d–e).

We used the Cyclone algorithm to characterize the distribution of neutrophils by phase of cell cycle[23]. As expected, most neutrophils were not actively cycling. However, cells in P1 (1.66% of all neutrophils) were skewed strongly toward G2M-like gene expression (Fig. 1f). This cluster was also distinguished by a more diverse per-cell transcriptome, with a higher number of detected genes, a higher number of transcripts, and greater transcriptional diversity as reflected in the lowest number of counts among the 500 most frequent features (Fig. 1g). Whereas we had pre-excluded granulocyte-monocyte progenitors (GMP) by mapping cells to ImmGen reference populations, we explored the possibility that P1 cells represented proliferation-competent immediate developmental precursors of neutrophils, recently described as pre-neutrophils (preNeus)[24]. Single-cell data characterizing preNeus have not yet been reported. We therefore obtained published bulk RNA-seq data[24] and calculated a gene set that best differentiated preNeus from GMP, immature and

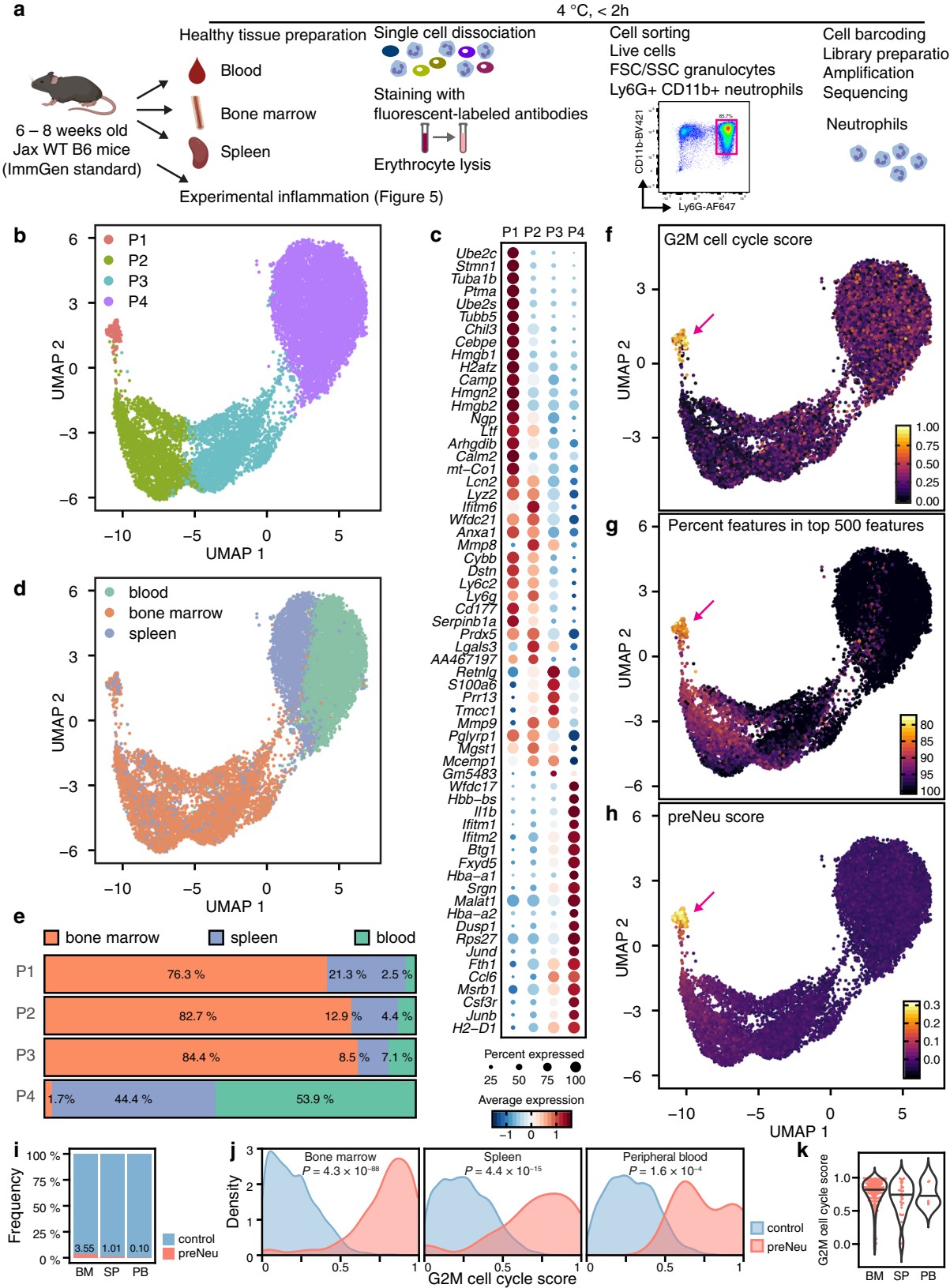

mature bone marrow neutrophils, and circulating neutrophils (Supplementary Fig. 4). We then calculated a gene module score for each cell in our dataset based on marker genes that were also robustly expressed in our dataset. This analysis revealed that P1 cells were most likely preNeus (Fig. 1h). The frequency of preNeus was highest in bone marrow (3.55%), followed by spleen

(1.01%) and blood (0.10%) (Fig. 1i). Although preNeus were detected in peripheral blood, their low abundance limited options for experimental examination. PreNeus exhibited a higher G2M cell cycle score compared to the remaining neutrophils (Fig. 1j), and the G2M cell cycle score was similar across tissues, consistent with their assigned identity (Fig. 1k).

**Fig. 1 Generating single-cell transcriptomes from neutrophils across multiple biological tissues. a** Overview of the experiment. Neutrophils were isolated from 6–8-week-old B6 mice from blood, bone marrow and spleen, stained for Ly6G and CD11b, and sorted, followed by droplet-based scRNA-seq using the 10X platform. Two independent experiments of healthy tissues were performed with $N = 3$ mice pooled for each tissue per experiment, totaling $N = 12{,}619$ cells, which were combined for analysis. **b** UMAP plot including all healthy neutrophils, partitioned exemplarily into four populations P1–P4. For smaller or larger numbers of populations, see Supplementary Fig. 3. **c** Marker gene expression in the 4-population model. Marker genes were identified by Wilcoxon Rank Sum test (two-tailed) using the Seurat function "FindAllMarkers" with standard settings; only genes with $\log_e$ fold change $\geq 0.25$ and Bonferroni adjusted $p$-value $\leq 0.05$ are shown. **d** UMAP embedding of all cells colored by tissue of origin. **e** Abundance of populations across organs. Neutrophils colorized by: G2M cell cycle score (**f**), percentage of features in top 500 features (**g**) and preNeu score (**h**). **i** Frequency of preNeu across tissues. **j** G2M cell cycle score in preNeu and remaining cells in each tissue. Unpaired t-test (two-tailed) between preNeu and all other neutrophils within each tissue. **k** G2M cell cycle score of preNeu across tissues. ANOVA followed by unpaired $t$-test was used to compare the G2M cell cycle score between preNeu and other neutrophils in each tissue.

Given the highly variable number of clusters obtained under a range of plausible UMAP assumptions, we sought alternate dimensional reduction strategies. We applied Monocle 3, another graph-based approach to identify cell trajectories in a low-dimensional space[25]. Aside from the small cluster identifiable as preNeus, Monocle revealed a continuous distribution of neutrophils across bone marrow, blood and spleen (Supplementary Fig. 5a). Broad correspondence between Monocle's pseudotime analysis and the clusters identified by UMAP was evident, but Monocle analysis did not integrate preNeus into the continuum, inducing us to adopt a diffusion map approach for subsequent analysis steps (Supplementary Fig. 5a).

**Neutrotime defines a continuous chronological spectrum of neutrophils.** The interconnectedness evident by Monocle, together with the relative instability of calling distinct clusters within the UMAP space, led us to consider whether neutrophils could be depicted more usefully through a continuous model. We therefore applied diffusion maps, a dimensional reduction method that orders cells based on transition probabilities and is sensitive to branches in the data[26–29]. Using this strategy, we found that neutrophils across all tissues could readily be mapped onto a single spectrum (Fig. 2a). Cells from bone marrow resided at one end of this continuum, while neutrophils in blood and spleen clustered at the opposite end, suggesting again a spectrum based on maturation (Fig. 2b). Neutrophils widely spread across this spectrum were observed in circulation, consistent with neutrophil release from healthy marrow at multiple stages of differentiation (Fig. 2b).

We tested diffusion mapping against UMAP dimensional reduction by mapping UMAP-defined populations P1–P4 onto neutrotime, confirming concordance between strategies (Fig. 2c). Considering the possibility that these clusters represented accumulation points in neutrophil differentiation, we analyzed cell abundance along the continuum (Fig. 2d). Most neutrophils were found at the mature end of the spectrum, with a second abundance cluster in the less mature middle and a third cluster at the very immature end, the latter largely in bone marrow and reflecting preNeus. We applied Hartigan's Dip Test to a cell-to-cell distance matrix, as well as to a cell ordering obtained from the diffusion map, confirming a multimodal distribution (Supplementary Fig. 5d). These results support the model that preNeus give rise to immature neutrophils that subsequently develop into mature neutrophils. Application of principal component analysis paired with a principal curve[30] to the primary scRNA-seq data yielded a similar distribution. Mapping neutrotime onto the principal curve and vice versa showed high convergence and a Spearman correlation of 0.955 (Supplementary Fig. 5c). Interestingly, however, 4 groups emerged in the abundance plot. The first 3 groups were largely found in bone marrow (Supplementary Fig. 5b), suggesting specific accumulation points in early neutrophil development. Spleen contained neutrophils at all

stages, but featured in particular an abundance of later-stage but still somewhat immature neutrophils, consistent with in vivo microscopy findings[31]. The large majority of neutrophils in the blood formed a single abundance peak at the highly mature end of the spectrum, cells that were rare in the marrow, indicating either that the most mature neutrophils are released rapidly into the blood or that the final stages of neutrotime-defined differentiation occur during release and/or in circulation (Fig. 2d).

To ensure that our analysis had not obscured developmental branch points, we employed Monocle 3, a machine learning strategy specialized in the identification of branching within scRNA-seq data[25]. Monocle disclosed no evidence for distinct trajectories of differentiation at the resolution provided by our dataset (Supplementary Fig. 5a).

Next, we sought to test whether this continuum reflected chronological order. We began with RNA velocity, a method that compares unspliced and spliced mRNA to provide an unambiguous time vector to transcriptomic data[32]. RNA velocity vectors indicated unidirectional progression from one end of the continuum to the other (Fig. 2e). To provide experimental confirmation of this temporal sequence, we obtained bulk RNA-seq data from developing Hoxa9-Lyz-GFP cells (male B6.129 (Cg)-Lyz2tm1.1Graf/Mmmh donor)[33]. These cells are wild-type mouse bone marrow cells reversibly immortalized at the myeloid cell progenitor stage by conditional overexpression of *Hoxa9*; when released from developmental blockade, they differentiate synchronously into mature myeloid cells. We examined expression of key genes along the continuum as HoxA9 cells became neutrophils. As expected, between 96 and 120 h, we observed decreasing expression of early-continuum genes such as *Camp*, and *Chil3*, as well as a decrease in the neutrophil survival factor *Serpinb1*, just as in our proposed maturational spectrum. We also noted increasing expression of late-continuum genes such as *Il1b*, *Ccl6* and the transcription factor *Junb*, again supporting the maturational sequence (Fig. 2f). These results are further illustrated by comparing the neutrotime score of cells ordered by rank in neutrotime, a visualization method that disclosed two discrete increments in neutrotime, corresponding to the transition from preNeus to immature neutrophils and from more mature marrow neutrophils to fully mature cells in circulation (Fig. 2g). Together, these data indicate that the neutrotime spectrum reflects chronological order and describes a single main sequence of neutrophil development across healthy bone marrow, blood and spleen.

**Neutrotime is defined by a structured gene expression program.** We sought to understand the transcriptional changes associated with progression along neutrotime. We identified the genes that showed the strongest positive and negative correlations with neutrotime, independent of source tissue. We found a total of 48 genes with Spearman correlation greater than 0.25 and 66

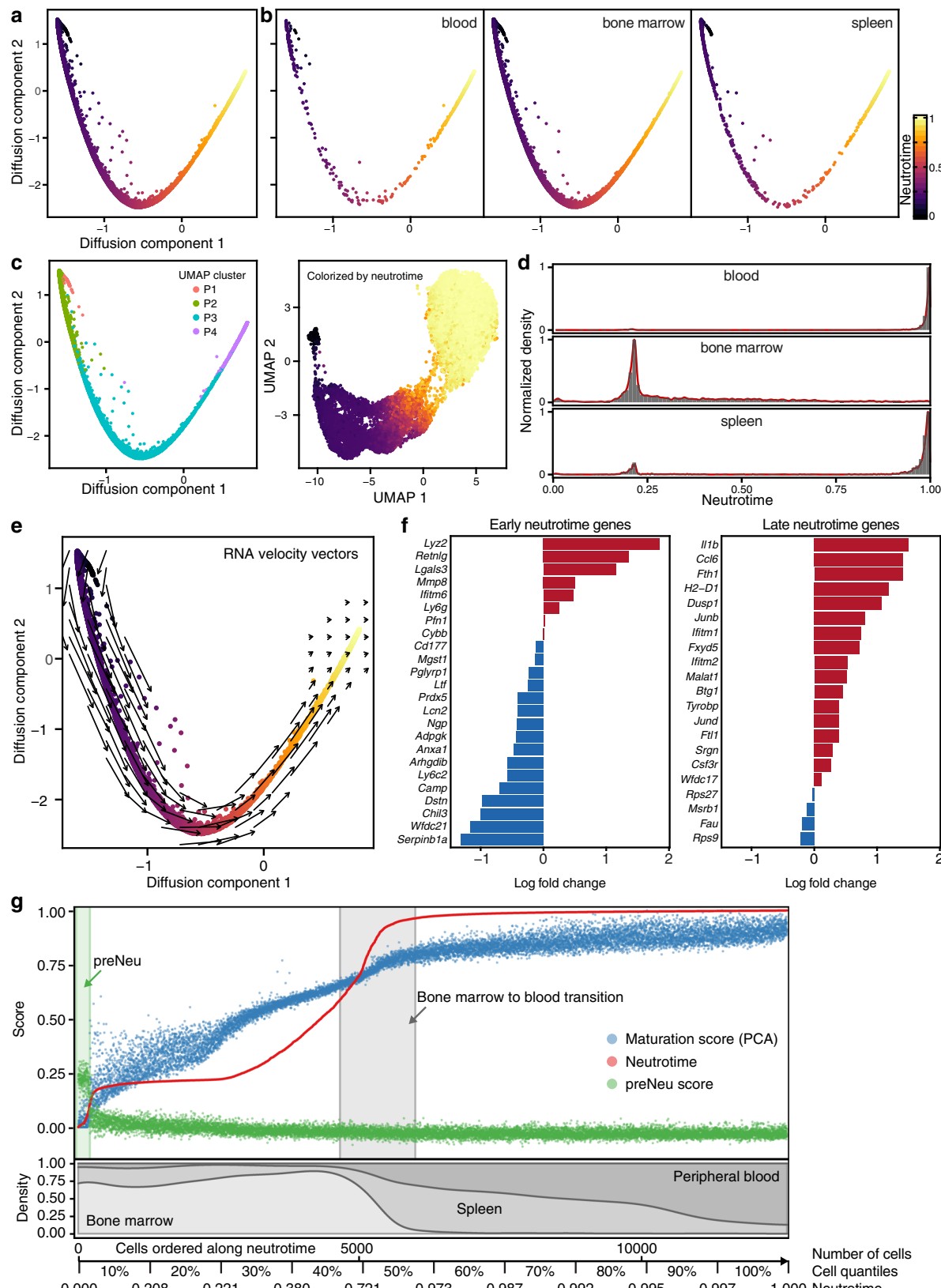

genes with Spearman correlation lower than −0.25 (Fig. 3a). We then plotted the expression of the top 50 genes that decrease with neutrotime and the top 50 genes that increase with neutrotime (Fig. 3b). Consistent with analysis through Monocle and diffusion map models, gene expression transitioned without discrete breakpoints from an early program to a late program profile

featuring de novo expression of a distinct gene set. Not all genes changed in a uniform direction over neutrotime; some, such as *Retnlg* encoding Resistin-like gamma precursor, *Mmp8* and *Mmp9* encoding the Matrix metalloproteinases 8 and 9, and *Mcemp1* encoding Mast Cell Expressed Membrane Protein 1, peaked between the poles of neutrotime (Fig. 3c). We further

**Fig. 2 Derivation of a neutrotime signature across biological compartments. a** Low-dimensional embedding of neutrophils from bone marrow, blood and spleen in the same space using a diffusion map-based approach. **b** Diffusion map for each tissue. **c** Convergence between UMAP populations P1–P4 and diffusion map ordering, as well as neutrotime representation on the UMAP embedding. **d** Relative cell density along neutrotime in each organ. **e** RNA velocity vector field on the neutrotime embedding. **f** Validation of the neutrotime gene signature in developing HoxA9 cells (log$_e$ fold change at 96 h and 120 h of differentiation). **g** Distribution of neutrotime (derived from diffusion map), maturation score (derived from principal component analysis, see Supplementary Fig. 5a–c) and preNeu score in cells ordered along neutrotime. Lower panel indicates the rolling density of cells in each compartment along neutrotime. The scale indicated below also serves as legend for other figures in which cells are ordered along neutrotime on the x axis.

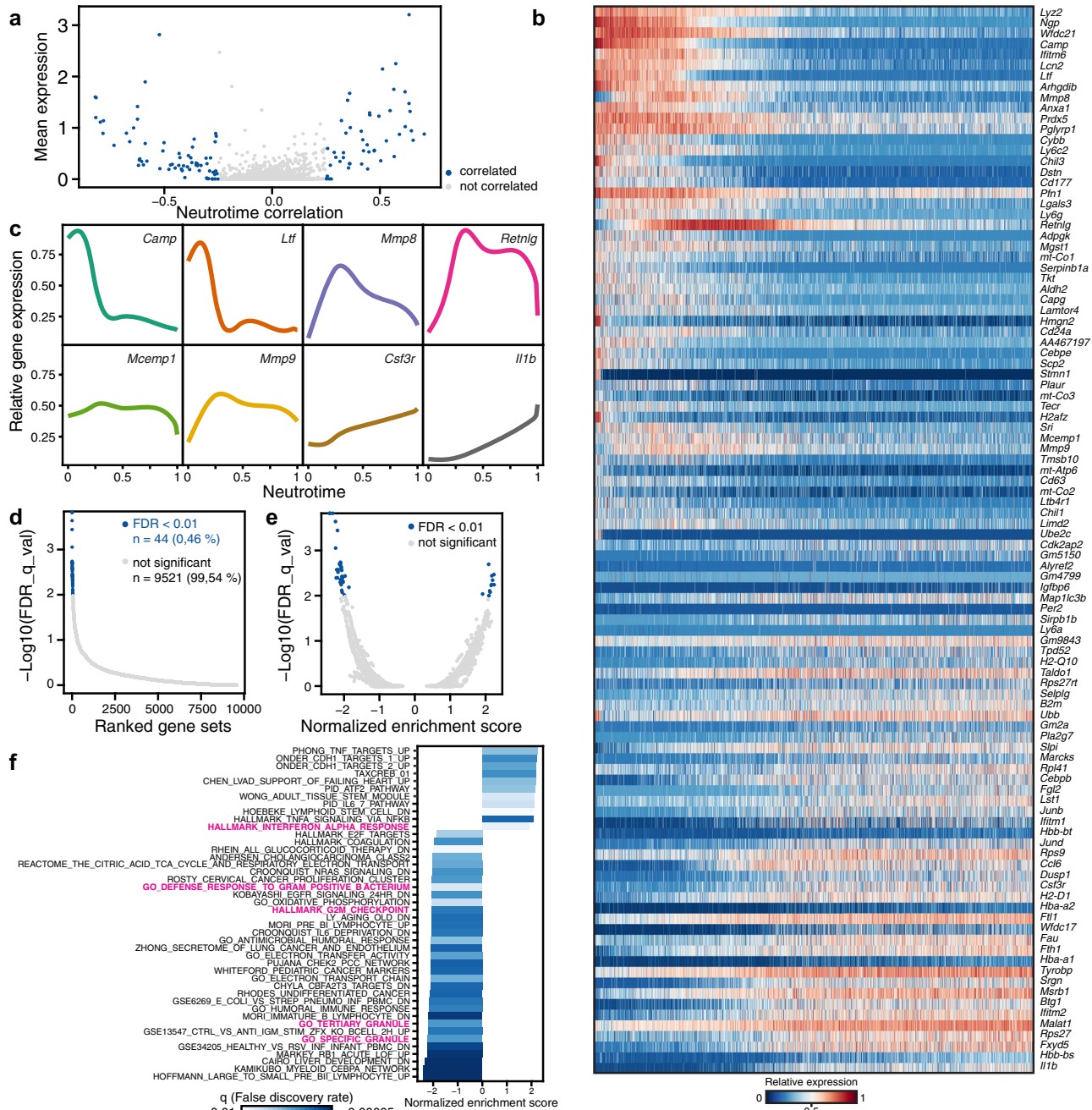

**Fig. 3 Gene expression along neutrotime. a** Gene expression correlation with neutrotime. Genes with Spearman correlation ≥ 0.25 (N = 48) and ≤ −0.25 (N = 66) are highlighted. **b** Gene expression heatmap of the top 50 positively and negatively correlated (Spearman) genes along neutrotime. **c** Gene expression profiles of select genes along neutrotime. **d** Ranked gene sets from Gene Set Enrichment Analysis (GSEA) versus false discovery rate. **e** Normalized enrichment score of GSEA gene sets versus false discovery rate. **f** Significantly enriched gene sets. Neutrophil-specific terms are highlighted in magenta. GSEA was performed as previously described.[76] In short, a normalized enrichment score was calculated by going through a list of genes based on their Spearman correlation with neutrotime as a pre-ranked list and calculating a running-sum statistic, which was then normalized for differences in the sizes of the gene sets that were looked at (normalized enrichment score). To account for multiple hypothesis testing, an FDR approach was used to maintain a defined level of significance, values < 0.01 were considered.

characterized neutrotime progression using gene annotation. Gene Ontology (GO) terms active early in neutrotime reflected anabolic functions, including metabolic processes, and defense response. Gene sets in late neutrotime were dominantly enriched in cellular responses to the environment, including immune responses, the cellular response to toxic substances, and gas transport (Supplementary Fig. 6). Gene Set Enrichment Analysis (GSEA) similarly indicated enrichment of granule loading with early neutrotime and effector processes with late neutrotime (Fig. 3d–f). Many genes in these signatures are common among actively proliferating cells (neutrophil-specific GSEA terms highlighted).

Gene expression may follow complex patterns. To identify these patterns in neutrotime, we performed k-means clustering on all variable genes ($n = 3322$) in an expression matrix of log normalized counts, where cells were ordered by neutrotime score. We chose $k = 10$ because this clustering gave a result in which no group consisted of fewer than five genes. 9 of 10 gene clusters were informative and followed a uniform distribution; the remaining cluster, cluster 10, consisted of genes with heterogeneous expression dynamics, supporting the number of clusters chosen (Supplementary Fig. 7). This analysis was able to separate genes with slow and fast expression dynamics, as well as genes differing in expression abundance. For example, cluster 2 (including *Camp*, *Ngp*, *Ltf*, *Lcn2*, *Ifitm6*, *Lyz2*, *Wfdc21*) and cluster 8 (including *Mmp8*), featured genes associated with secondary granules[22] and were both associated with early neutrotime. Cluster 2 peaked earlier and displayed a sharper decline in expression, whereas genes belonging to cluster 8 were present further along neutrotime. Thus, the application of the neutrotime paradigm identifies patterns in gene expression dynamics as neutrophils mature (Supplementary Fig. 7).

**Neutrotime varies across tissues**. Analysis of the relative abundance of cells along neutrotime had shown that early neutrotime is dominated by bone marrow and late neutrotime by blood and spleen (Fig. 2g). We thus investigated the degree of transcriptional heterogeneity in the dataset introduced by source tissue versus neutrotime. We binned together cells using a continuous increase in neutrotime (0.00001), smaller than the smallest difference of neutrotime between two cells, until at least 5 cells per tissue and at least 30 cells total were represented in each bin. This approach yielded a continuous collection of cells in 76 bins. We then calculated the average gene expression of neutrophils in each respective bin, separately for each tissue. Visualization of the core neutrotime transcripts disclosed a remarkable similarity among expression profiles of key neutrotime genes, irrespective of tissue (Supplementary Fig. 8). Differences were observed only in few genes, such as *Ly6g* (almost no expression in peripheral blood). *Hbb-bs*, encoding beta-globin, was detectable primarily in neutrophils from peripheral blood, potentially representing cell-free RNA from erythrocytes, as was recently described to be common in single-cell experiments[34].

**Expression of type I interferon-related transcripts varies across neutrotime**. Neutrophils are exquisitely responsive to type I interferon, as reflected in the prominent transcriptomic impact of in vivo administration of this cytokine[35] (Fig. 4a). Neutrophils exhibiting transcriptional evidence of interferon response have been identified in models of cancer and inflammation, although not previously in healthy mice[16,36]. Interestingly, neutrotime-defined maturation was prominently associated with a GSEA signature for type I interferon response (Fig. 3f). We thus examined expression of individual genes associated with type I interferon response along neutrotime (Fig. 4b). *Ifitm2* and *Ifitm1*

increased progressively with neutrotime, *Ifitm3* was expressed throughout neutrotime, and *Ifitm6* was expressed preferentially early in neutrotime (Fig. 4c). By contrast, genes associated with type II interferon response were low in expression and exhibited no consistent pattern (Fig. 4d). An attempt to identify a discrete group of neutrophils by clustering on expression of type I response genes such as *Ifit1*, *Ifit3*, *Isg15*, and *Ifitm3* failed to identify a subset of cells distinct from the neutrotime spectrum (Fig. 4e). Instead, genes associated with type I interferon response in neutrophils peaked at different points along neutrotime (Fig. 4c, f), suggesting that the expression of interferon target genes in healthy neutrophils is a dynamic process that evolves with maturation. Whether this gene expression pattern arises through direct interferon exposure or by other means, and how interferon-related transcripts change neutrophil function, remains to be determined.

**Neutrotime can be validated in published mouse bulk sequence data**. To test neutrotime against published data, we used public bulk RNA-seq data from proliferation-competent, lineage-committed preNeus (Lin⁻ CD115⁻ SiglecF⁻ CD11b⁺ Gr1⁺ cKit$^{int}$ CXCR4⁺), immature bone marrow neutrophils (Lin⁻ CD115⁻ SiglecF⁻ CD11b⁺ Gr1⁺ cKit$^{low/neg}$ CXCR4$^{−/low}$ Ly6G$^{low/+}$ CXCR2⁻), mature bone marrow neutrophils (Lin⁻ CD115⁻ SiglecF⁻ CD11b⁺ Gr1⁺ cKit$^{low/neg}$ CXCR4$^{−/low}$ Ly6G⁺ CXCR2⁺), and mature blood neutrophils (Lin⁻ CD115⁻ SiglecF⁻ CD11b⁺ Gr1⁺ cKit$^{low/neg}$ CXCR4$^{−/low}$ Ly6G⁺ CXCR2⁺) to define the location of these populations along the neutrotime spectrum[24]. Diffusion maps as used to derive neutrotime create a low-dimensional embedding of the analyzed cells wherein the contribution of single genes cannot be directly defined. We therefore generated a simplified version of neutrotime using a subset of the most informative genes. We first selected genes profiled in Fig. 3a (Spearman correlation). We next constructed a cell trajectory within a UMAP embedding created through the Monocle pipeline and ran Moran's I, a measure of spatial autocorrelation, to identify variable factors within a two-dimensional space[37]. We then selected genes that displayed high Spearman correlation with neutrotime, a high rank in Moran's I, and the highest expression abundance, retaining only the 100 most informative genes (Fig. 5a). Each gene was categorized into three groups based on negative, positive or little correlation to neutrotime. We then calculated a simplified neutrotime-S score for each cell in our original dataset directly from gene expression space (methods). We tested neutrotime-S against the full neutrotime and found a Spearman correlation coefficient of 0.905 (Fig. 5b). We could then calculate a neutrotime-S score for each population from published data. PreNeus had the lowest neutrotime-S, followed by immature bone marrow neutrophils, mature bone marrow neutrophils, and finally blood neutrophils, supporting the neutrotime paradigm (Fig. 5c). PreNeus and immature marrow neutrophils localized closely together by neutrotime-S, as did mature marrow neutrophils and mature blood neutrophils; however, a large gap separated immature and mature populations, highlighting the marked shift in gene expression that characterizes this transition (Fig. 5c). Mature neutrophils in the bone marrow and mature neutrophils in the blood were separated by a small but distinct increment in neutrotime-S (from 0.9450 to 0.9963). Differential expression analysis identified 981 differentially expressed genes that characterized this transition, highlighting that blood neutrophils are distinct from even the most mature marrow neutrophils (Fig. 5d). Many of the core neutrotime genes were among the differentially expressed genes, including upregulation of *Il1b* and downregulation of *Lcn2*, *Retnlg* and *Ly6g* in peripheral blood compared with marrow.

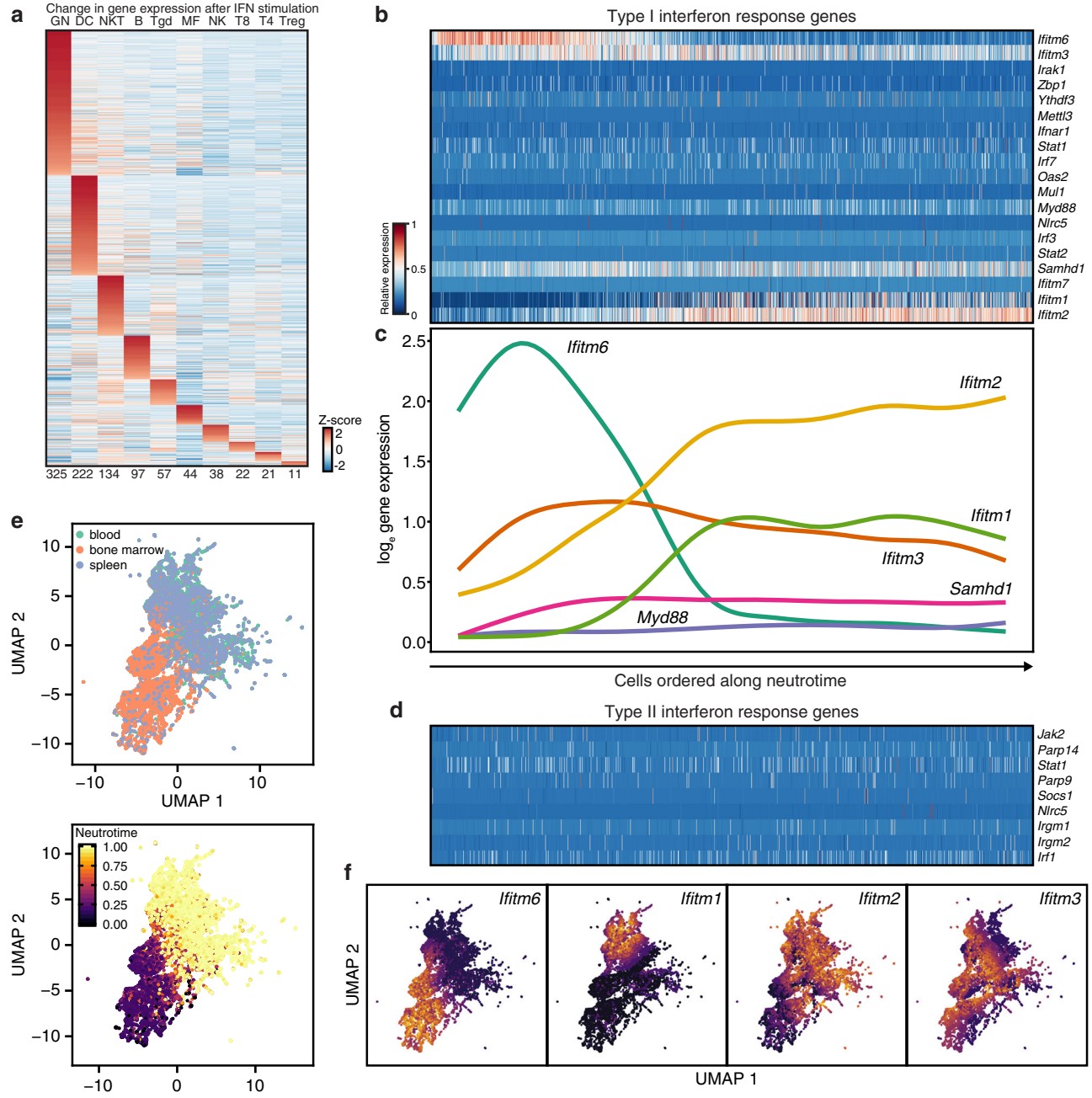

**Fig. 4 Expression of interferon-related transcripts in neutrophils. a** Response of 10 ImmGen populations to in vivo interferon α administration highlights marked transcriptional shift in neutrophils. **b**, **c** Expression of transcripts associated with type I interferon response along neutrotime. Cells are ordered along the x axis as in Fig. 2g. **d** Expression of transcripts associated with type II interferon response along neutrotime. **e** UMAP clustering of neutrophils based exclusively on transcripts associated with type I interferon response. **f** Expression of transcripts associated with type I interferon response mapped onto the UMAP embedding.

**The neutrotime signature can be detected in human neutrophils.** To test whether neutrotime translated to human neutrophils, we obtained scRNA-seq data from bone marrow neutrophils in the Human Cell Atlas[38], available through SeuratData[39]. We manually removed non-neutrophils from the dataset through dimensionality reduction and clustering based on expression of marker genes (e.g., *GZMB* and *GNLY* for NK cells). Next, we queried genes with Spearman correlation of neutrotime ≤ 0.25 (early neutrotime genes) and ≥ 0.25 (late neutrotime genes) for human 1:1 orthologs with high confidence according to ENSEMBL version 100. A score for early and late neutrotime was then calculated for each cell and plotted on a UMAP embedding

of enriched neutrophils (Fig. 5d–f). These results confirm that human neutrophils exhibit a gene expression pattern broadly consistent with the neutrotime paradigm identified in mice.

**Serial expression of transcription factors through neutrotime.** Progression along neutrotime was accompanied by global changes in gene expression. We tested the possibility that these changes reflected coordinate gene regulation by specific transcription factors (TFs). Indeed, TF expression evolved over the course of neutrotime (Fig. 6a, b). To evaluate TF function, we applied ChIP-X Enrichment Analysis Version 3 (ChEA3), a

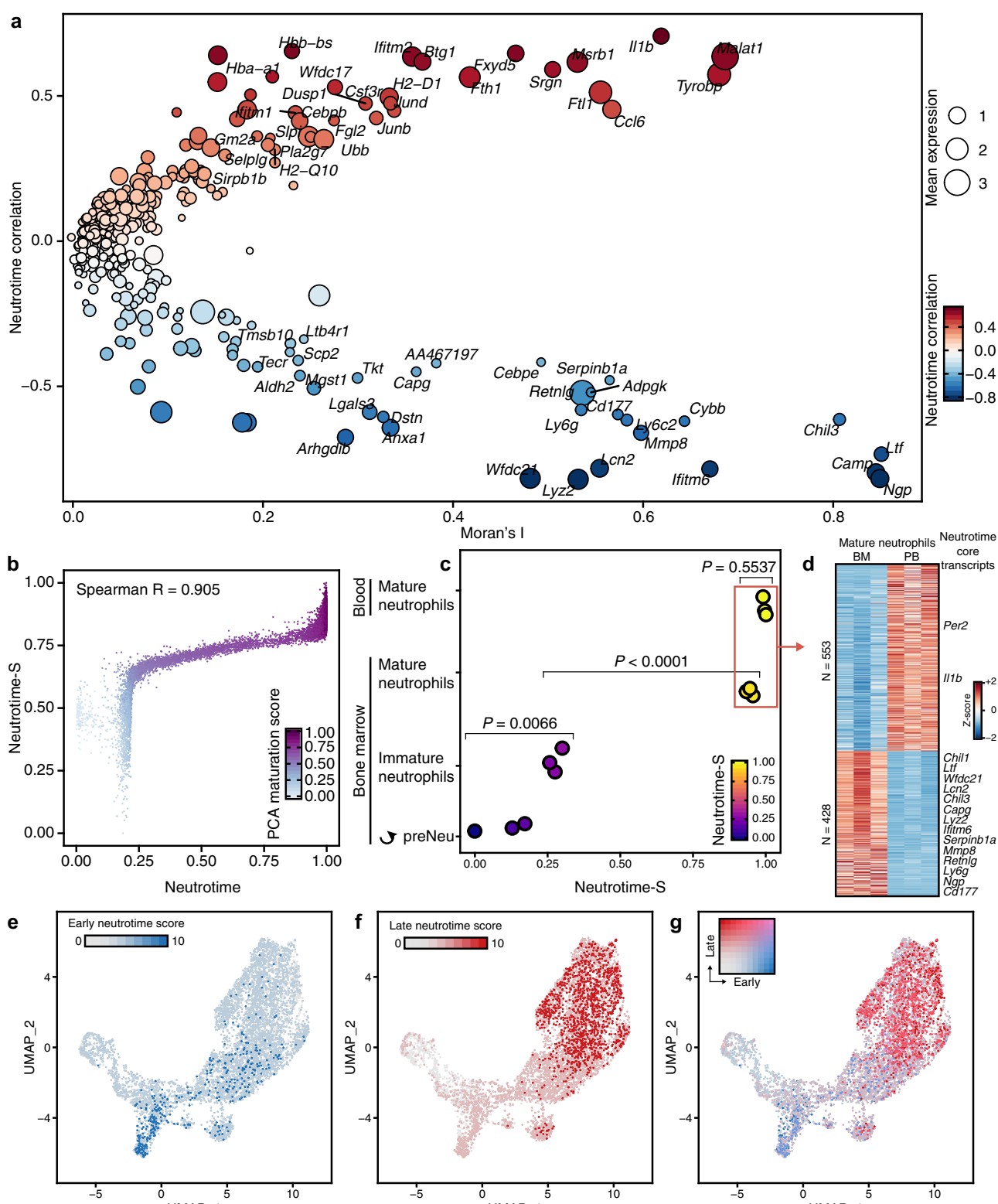

**Fig. 5 Validation of the neutrotime model. a** Spatial autocorrelation (Moran's I) of genes calculated from a separately obtained dimensionality reduction using Monocle 3 plotted versus genes correlated with neutrotime. **b** Calculation of a simplified version of neutrotime directly from gene expression space yields high convergence with neutrotime. *R* indicates the Spearman's rank correlation coefficient. **c** Integration of bulk RNA-seq data from developing and mature neutrophils from Evrard et al. (GEO:GSE109467) onto neutrotime-S. One-way ANOVA (*P* < 0.0001) followed by Tukey's multiple comparisons test. **d** Heatmap detailing differentially expressed genes between mature bone marrow neutrophils and blood neutrophils with Benjamin & Hochberg adjusted *p*-value (corresponding to FDR) ≤ 0.05 and log₂ fold change ≥ 1. Several core neutrotime transcripts are highlighted. **e**–**g** scRNA-Seq data were obtained from the Human Cell Atlas, and cells belonging to the neutrophil lineage (*N* = 7049) were clustered together. Expression of an early (**e**) and late (**f**) neutrotime signature in neutrophils. **g** combined representation of early and late neutrotime scores in the Human Cell Atlas confirms the trajectory.

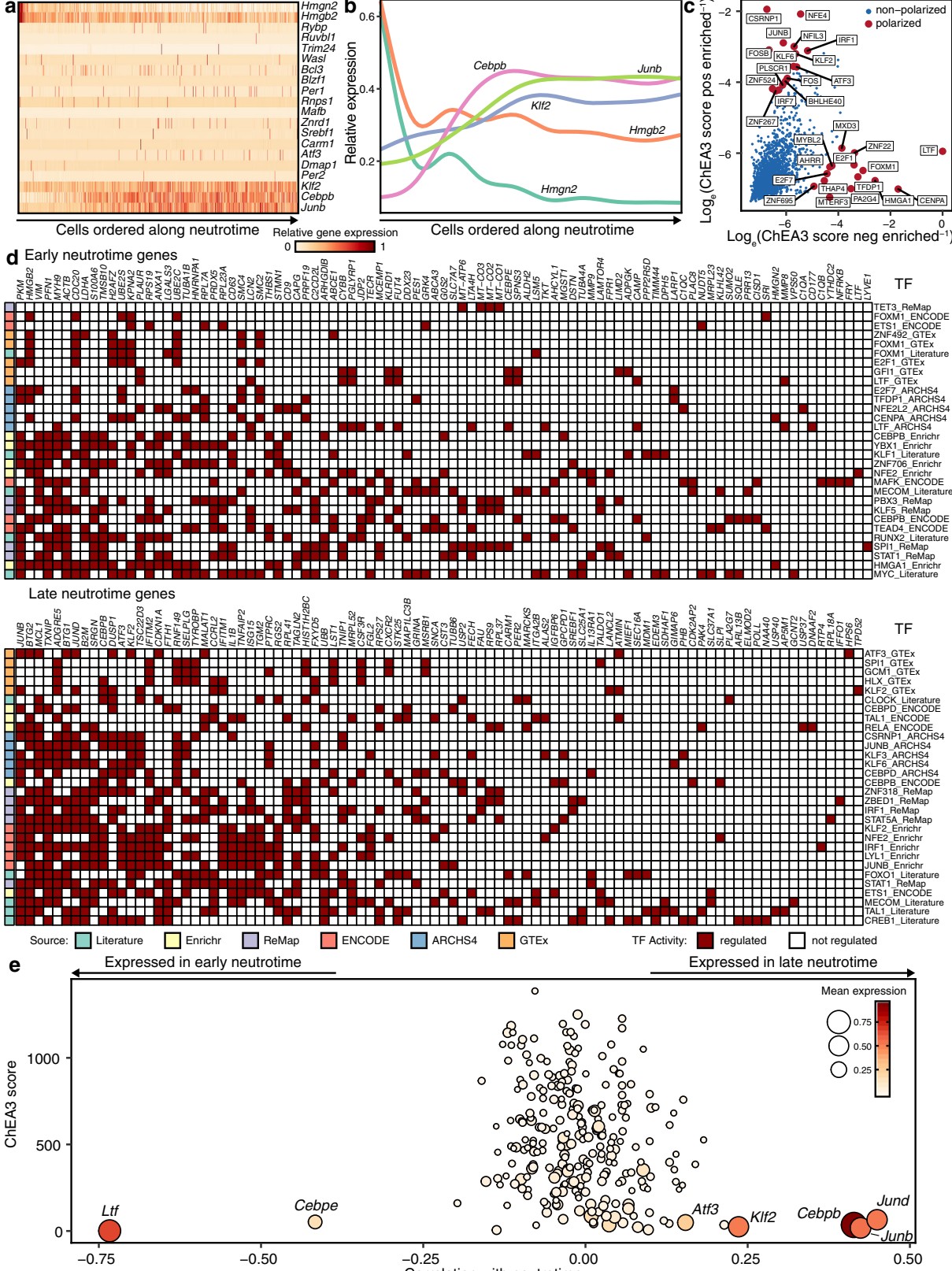

**Fig. 6 Transcriptional regulation of neutrotime. a**, **b** Expression profiles of highly abundant transcription factors along neutrotime. **c** Inferred transcriptional regulators of early and late neutrotime via ChEA3. **d** Transcriptional orchestration of early and late neutrotime. **e** Inferred regulatory activity versus transcription factor expression along neutrotime. Cells are ordered along neutrotime in **a** and **b** with the same x axis scale as in Fig. 2g. As ChEA3 contains libraries from multiple species, only the corresponding human protein symbols are shown in **c** and **d**.

methodology that seeks the signature of specific TFs in the patterns of differentially expressed genes, based on mouse and human data from ENCODE, ReMap, ARCHS4, GTEx, Enrichr, and curated results from the literature[40]. Strong evidence emerged that different TFs are active in early vs. late neutrotime (Fig. 6c). Plotting TF activity by ChEA3 against regulated genes, no single TF emerged as likely to be solely responsible for evolution across neutrotime; rather, different modules of the transcriptional program are likely driven by distinct TFs (Fig. 6d). Combining raw RNA-seq data with ChEA3 scores, we identified a remarkably restricted set of TFs that both varied with neutrotime and exhibited strong ChEA3 scores (Fig. 6e). These results were robust to the number of genes considered for the ChEA3 calculation, as extending the list to the top 200 or top 300 genes highlighted the same TFs (Supplementary Fig. 9). Specifically, lactoferrin (*Ltf*, a secondary granule protein but also a known TF) and *Cepbe* were strongly active in early in neutrotime. By contrast, later neutrotime was characterized by the activity of TFs encoded by *Atf3*, *Klf2*, *Cepbb*, *Junb* and *Jund*. These results are consistent with available functional data, as illustrated by reciprocal expression of CCAAT enhancer binding protein (C/EBP) family members C/EBPε and C/EBPβ in early and late neutrotime[41]. Mice lacking C/EBPε display interrupted neutrophil maturation, with accumulation of neutrophil precursors and an absence of morphologically and functionally normal neutrophils in blood[42]. By contrast, C/EBPβ facilitates cytokine production and protection from apoptosis, especially in the inflamed milieu, capacities befitting late neutrotime[43,44]. Note that C/EBPα, another member of this family essential for neutrophil development, did not emerge from this analysis, consistent with fulfillment of its role earlier in myelopoiesis, preceding the preNeu stage[45].

**Inflammation induces polarization of neutrophils beginning from neutrotime.** To understand how gene expression varies with inflammation, we sorted neutrophils from mice undergoing three established models of sterile inflammation: K/BxN serum-induced arthritis (blood and joint lavage 7 days after i.p. administration of arthritogenic serum); IL-1β pneumonitis (lung lavage 3 h after i.n. instillation of 25 ng IL-1β); and IL-1β peritonitis (peritoneal lavage 3 h after i.p. 25 ng IL-1β) (Fig. 7a)[6]. Using the same filtering thresholds as for healthy neutrophils, high-quality scRNA-seq data could be obtained from 801, 2791, 287, and 367 cells from the four compartments, respectively (Supplementary Fig. 1f). Experiments were performed together with one of the studies of healthy neutrophils ($n = 506$ cells), with simultaneous 10X analysis by hashtagging to mark condition of origin, facilitating direct comparison across all groups (Supplementary Fig. 1c–f). Healthy and inflamed neutrophils were then analyzed together. Principal component analysis revealed that cells from inflamed tissues polarized in two directions, one for acute IL-1β-induced inflammation (lung or peritoneum) and the other for subacute K/BxN arthritis (Fig. 7b).

By neutrotime-S, K/BxN blood neutrophils were less mature than healthy blood neutrophils $0.585 \pm 0.003$ vs $0.639 \pm 0.003$ (mean ± standard error of the mean), suggesting early release of neutrophils in response to tissue inflammation. By contrast, neutrophils in the joint were more advanced than those in healthy or arthritic blood ($0.647 \pm 0.001$, mean ± standard error of the mean), while those from inflamed lung and peritoneum were relatively immature ($0.611 \pm 0.005$ and $0.584 \pm 0.005$, mean ± standard error of the mean), potentially suggesting recruitment of less mature neutrophils to acutely inflamed tissues (Fig. 7c). Transcriptional similarity distinguished patterns evident across all three inflammatory conditions and also patterns that

distinguished acute (lung/peritoneum) from subacute (joint) inflammation (Fig. 7d). By contrast, neutrophils from healthy and arthritic blood resembled each other, differing in only 39 transcripts at FDR < 0.01 and a $\log_e$ fold change of at least 0.25, in line with the joint focus of inflammation in the K/BxN model (Fig. 7d). Among these, circulating neutrophils from arthritic animals displayed greater expression of *Retnlg*, *Lcn2* and *Ly6g*, genes associated with relative immaturity. For a detailed analysis of gene expression changes in all inflammatory conditions see Supplementary Data 1.

Projecting neutrophils onto a diffusion map confirmed that healthy and arthritic blood neutrophils clustered together, and that neutrophils from lung and peritoneum diverged from those harvested from arthritic joints (Supplementary Fig. 10a–c). Similar results were obtained in the first two principal components (Fig. 7b, e). Comparing all three inflamed tissues to healthy blood, we observed a shared set of up- and down-regulated genes (Fig. 7d and Supplementary Fig. 10d). Genes with upregulated expression included the pattern recognition co-factor *Cd14*; the chemokines *Cxcl2*, *Ccl4, and Ccl3*; *Irg1*, encoding an enzyme involved in synthesis of the antibacterial factor itaconic acid; and ferritin heavy chain (*Fth1*). Antagonists of IL-1 signaling, *Il1r2* and *Il1rn*, were also upregulated. Using the same threshold ($\log_e$ fold change 0.5), genes downregulated across inflammatory conditions included *Tsc22d3*, *Rsrp1*, *Tmsb4x*, *Myadm* and *Tmcc1* (Supplementary Fig. 10e). *Retnlg*, encoding Resistin-like gamma precursor and *F630028O10Rik*, an uncharacterized protein, were downregulated specifically in neutrophils from inflamed joints. *Mmp9*, *Rgs2* and *Zfp36l2* were downregulated in both IL-1β-induced conditions, while *Lyz2* and *Lst1* were downregulated in IL-1β-induced pneumonitis but not peritonitis. No genes exclusively downregulated in IL-1β-induced peritonitis passed the threshold. Thus, neutrophils recruited to inflamed tissues exhibited substantial overlap in gene expression, notwithstanding differences in trigger and time course.

Differences between inflamed tissues were also detected. In particular, neutrophils from the joint differed from those harvested from lung and peritoneum (Fig. 7b, e). For example, neutrophils from pneumonitis and peritonitis (as well as healthy and arthritic blood) exhibited a predominance of *Cxcr2*, encoding the receptor for key neutrophil recruitment chemokines KC (CXCL1) and MIP2 (CXCL2). By contrast, neutrophils from inflamed joints were skewed toward expression of *Cxcr4*, a receptor for CXCL12 that enables return of aged neutrophils to the marrow but also retention of neutrophils within inflamed tissues[46,47] (Fig. 7f). Intriguingly, IL-1β-inflamed lung and peritoneum differed from each other in 224 transcripts at FDR < 0.01 and a $\log_e$ fold change of at least 0.25, including overexpression in lung neutrophils of *Plaur*, *Wnk1*, *Ccr1*, *Cd300lf*, *Cd33*, *Osm*, *Nfil3*, *Id2*, *Rnasel*, *Mt1 and Crispld2* and overexpression in peritoneal neutrophils of *Klf2*, *Ehd1*, *Tnfaip2*, *Marcks*, *Vasp*, *Nfkbiz*, *Rel*, *Dusp2*, *Ikbke*, *Tnip1*, *Bcl3*, *Ebi3*, *Rnd12810474O19*, *Rik*, *Batf*, *Cxcl10*, *Nr4a1*, *Ncf1*, and *Chil1* (Fig. 7d). Thus, gene expression among neutrophils recruited to inflamed tissues varies with site of recruitment, even when elicited by an identical stimulus over an identical interval.

**Associations between transcript and protein expression varies with context.** Transcript abundance bears an inconsistent relationship to the level of a protein expressed. To assess this relationship within neutrotime genes, we selected 8 genes that exhibited dynamic transcriptional regulation along neutrotime and with inflammation (*Cd9*, *Cd14*, *Cd53*, *Cd63*, *Itgam*, *Cxcr2*, *Cxcr4*, *Ly6g*) and compared transcript abundance from our scRNA-seq data with mean surface expression as measured by

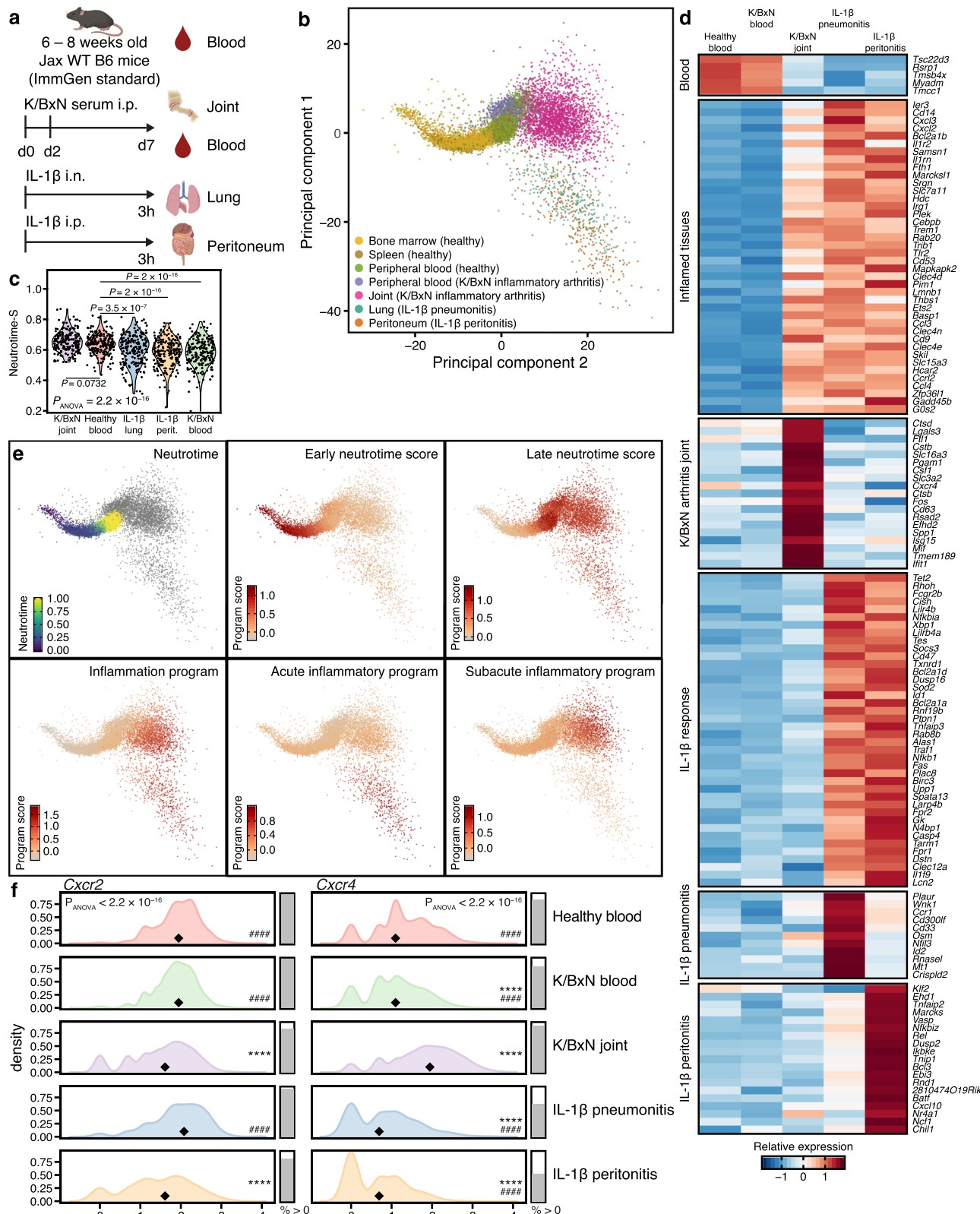

flow cytometry, expressed as change with respect to healthy blood neutrophils. As expected, the relationship between transcript and protein proved highly variable (Supplementary Fig. 11). This result illustrates the importance of caution in the extrapolation of transcriptional phenotype to cell function and highlights the strength of the transcript-only approach to cell ontology as realized in neutrotime.

**Inflammation drives new transcriptional activity in neutrophils.** We observed marked changes in transcription factor expression across inflammatory conditions (Fig. 8a). *Hmgb2*, encoding a DNA-interacting protein with antimicrobial properties, was largely restricted to blood neutrophils[48]. Neutrophils from acutely inflamed lung and peritoneum upregulated *Xbp1* and *Nfkb1*, implicated in neutrophil effector responses and their

**Fig. 7 Neutrophils in acute and subacute inflammation. a** Overview of the experimental models. **b** Combined principal component analysis of healthy and inflamed neutrophils highlights a divergence between acute IL-1β-induced and subacute (K/BxN serum transfer arthritis) inflammation. $N = 17,424$ cells from $N = 3$ mice per inflamed condition total and $N = 3$ mice per healthy tissue and per experiment (two independent experiments). **c** Differences in neutrotime-S between inflamed compartments. ANOVA followed by Dunnett's multiple comparison test (two-tailed) compared to healthy blood. **d** Heatmap summarizing shared and stimulus-specific gene expression changes in neutrophils. As for Fig. 1c, marker genes were identified by Wilcoxon Rank Sum test (two-tailed) using the Seurat function "FindAllMarkers" with standard settings; only genes with $\log_e$ fold change $\geq 0.5$ compared to healthy blood and Bonferroni adjusted $p$-value $\leq 0.05$ were considered. For comparisons that examined changes between multiple groups compared to healthy blood (e.g. "Inflamed tissues"), as most conservative approach, the highest adjusted $p$-value was chosen for each gene. **e** Original neutrotime score in healthy cells, early and late neutrotime scores in healthy and inflamed cells highlights that the late neutrotime program stays active throughout inflammation. Lower panels highlight different inflammatory programs in inflamed cells. **f** Antagonistic expression of *Cxcr2* and *Cxcr4* in acute versus subacute inflammation: distribution of *Cxcr2* and *Cxcr4* expression ($\log_e$ normalized expression) in healthy and inflamed tissues. Diamonds depict the median expression value and filled bars on the right illustrate the percentage of cells with non-zero expression. ANOVA was performed on the counts followed by Dunnett's multiple comparison test (two-tailed). Percentage of cells with non-zero expression shown as descriptive statistic. ****$P < 0.0001$ compared to healthy blood; ####$P < 0.0001$ compared to K/BxN joint.

control[49,50]. Lung and peritoneum differed in the expression of TFs including *Nfil3*, implicated in the control of IL-1β and TNF production by myeloid cells[51]. Neutrophils from arthritic joints expressed *Cebpb*, *Atf3*, and *Fos*. The late-neutrotime gene *Cebpb* is the hallmark TF of emergency / reactive granulopoiesis and modulates cytokine production[44,52]. *Egr1* and *Fos* are suppressed by the key neutrophil-differentiation transcription factor GFI-1 in developing neutrophils; later expression may be involved in production of mediators including IL-1β[53,54]. *Atf3* participates in neutrophil migration[55]. Examining transcription factor regulatory activity as inferred through ChEA3 again suggested differential participation of TFs in orchestrating homeostatic and inflamed gene expression programs (Fig. 8b–c). Plotting transcription factor activity by ChEA3 against transcript expression, we identified TFs associated with acute inflammation in response to IL-1β as well as K/BxN immune-complex arthritis, most prominently *Cebpb* expression and function in both inflamed conditions (Fig. 8d). This variability among transcriptional programs, driven by distinct sets of TFs, highlights the dynamic nature of gene regulation in mature neutrophils as they emerge from a single neutrotime continuum to assume effector roles in the tissues.

## Discussion

Neutrophils represent an evolutionarily ancient component of innate immune defense[56]. Despite considerable morphological homogeneity, neutrophils are phenotypically diverse, and it has proven useful to group together neutrophils that share specific features; examples include low-density granulocytes, N2 / suppressor neutrophils, and pro-angiogenic neutrophils[57–59]. However, the ontological relationship among these populations, sometimes termed neutrophil subtypes, is unknown. In particular, it is unknown whether there are different kinds of neutrophils, as NK cells and CD4 + T cells are different kinds of lymphocytes, or whether phenotypic variation among neutrophils is better understood as differential maturation and activation within a single cell type.

To define the underlying organization of the neutrophil population, we applied single-cell transcriptomics, a method that has emerged as a powerful tool to study immune cell heterogeneity[13]. We studied more than 17,000 sorted neutrophils from bone marrow, blood and spleen of healthy mice, from inflammatory exudates in peritoneum and lung, and from both blood and joint in sterile subacute arthritis. We ordered these cross-sectional profiles to expose relationships among the phenotypes observed. Using distinct modeling strategies, we observed a consistent organizational logic: the differential transcriptional signatures of neutrophils reflect a single continuum across tissues, without major branch points. This continuum converges with

chronological order, as determined through RNA velocity and from neutrophils differentiated in vitro, leading us to term this trajectory neutrotime. Our data therefore define a single main sequence of mouse neutrophil development.

This overall population structure resembles that suggested by Evrard et al.[24], whereby neutrophils progress from proliferation-competent committed precursors (preNeus) through a non-proliferating immature stage to mature neutrophils. Cell abundance data further underscore this preNeu/immature/mature division, identifying accumulation points in neutrophil development corresponding to preNeus, nearly mature neutrophils awaiting transit into blood, and mature circulating neutrophils. Although neutrophils all along this spectrum can be identified in marrow, spleen and blood, the distribution changes markedly with location. In healthy mice, preNeus reside predominantly in the marrow, though some are encountered in spleen (a known hematopoietic organ) and rare preNeus appear in the blood. Spleen neutrophils predominantly reflect the mature end of the spectrum, though a substantial fraction are less mature than circulating neutrophils, consistent with in vivo microscopy data[31]. Blood neutrophils in healthy mice overwhelmingly occupy the most mature end of the spectrum. Interestingly, a sharp increase in neutrotime accompanies the transition from marrow to blood, suggesting that this transit event or other factors associated with circulation participate in final neutrophil maturation.

Attention to variation of TF expression and function with neutrotime demonstrates the extent to which neutrophil development is transcriptionally dynamic, including a previously unappreciated transcriptional program of terminal neutrophil differentiation. Early neutrotime is under the control of TFs such as C/EBPε, consistent with the requirement for this TF in the development of normal mature neutrophils in mice and humans[42,60]. *Ltf* encoding lactoferrin was also prominent in early neutrotime. A secondary granule protein, lactoferrin is also a well-recognized TF, binding with high affinity to specific DNA motifs[61,62]. The protein sequence of lactoferrin includes a nuclear localization signal that is highly conserved across mice and humans, consistent with important transcriptional activity[63]. Interestingly, mice lacking lactoferrin display normal early neutrophil development, although the respiratory burst is impaired in mature neutrophils, suggesting redundancy in the control of early neutrotime[64].

Later steps in neutrotime reflect the activity of a distinct group of TFs including ATF3, KLF2, C/EBPβ, JUNB, and JUND. This evidence for reciprocal roles for C/EBPε (early neutrotime) and C/EBPβ (late neutrotime) concord with focused studies of these TFs[42,60]. ATF3 facilitates the capacity of neutrophils to migrate into inflamed tissues[55]. KLF2 is implicated in neutrophil resistance to apoptosis[65]. JUNB plays a key role in enhancing

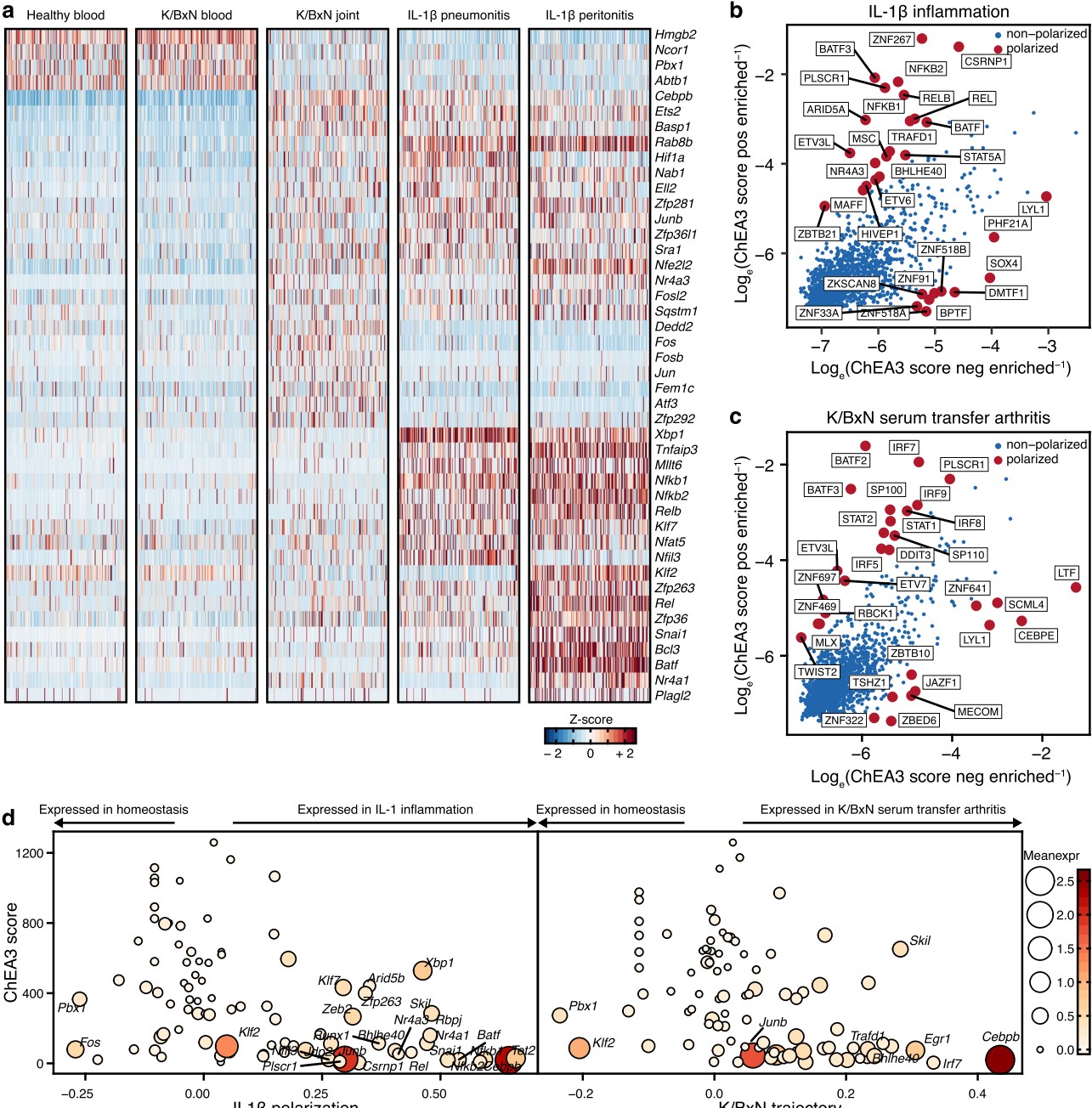

**Fig. 8 Transcriptional regulation of acute and subacute inflammation. a** Overview of differentially expressed transcription factors in healthy blood and inflamed compartments. As in Fig. 1c, marker genes for each condition were identified by Wilcoxon Rank Sum test (two-tailed) using the Seurat function "FindAllMarkers" with standard settings; only genes with log$_e$ fold change between conditions ≥ 0.25 and Bonferroni adjusted $p$ value ≤ 0.05 were considered. The list of marker genes was subsetted to transcription factors to display differentially expressed transcription factors between conditions. Cells were randomly downsampled to 200 cells per condition for plotting only. **b, c** Inferred regulatory activity of TFs in IL-1β-induced and K/BxN-induced inflammation. As ChEA3 contains libraries from multiple species, only the corresponding human protein symbols are shown. **d** Inferred activity versus actual expression of TFs in IL-1β polarization and along the K/BxN trajectory in the diffusion map.

neutrophil effector potency as expression of the myeloid differentiation TF PU.1 fades[66]. In macrophages, JUNB promotes transcription of *Il1b*, a hallmark of late neutrotime[67]. JUND is similarly implicated in the upregulation of genes related to the production and processing of IL-1β[68]. These TFs were identified by an algorithm that sought TFs exhibiting both transcriptional variation with neutrotime and evidence of function as disclosed by ChEA3. Our data do not exclude the possibility that other TFs may also play key roles in neutrotime, in particular for TFs that may not be regulated at the transcriptional level. The contribution

of each TF to neutrophil development and function will require further experimental dissection. Ng, Ostuni and Hidalgo propose a key role for developmentally-conditioned epigenetic differences, a fertile hypothesis that remains to be tested[3].

To understand the transcriptional correlates of recruitment to inflamed tissues, we characterized neutrophils from IL-1β peritonitis, IL-1β pneumonitis, and immune-complex arthritis. Gene expression in neutrophils from all 3 inflamed sites differed strikingly from that in healthy cells, confirming that circulating neutrophils remain transcriptionally dynamic. Importantly,

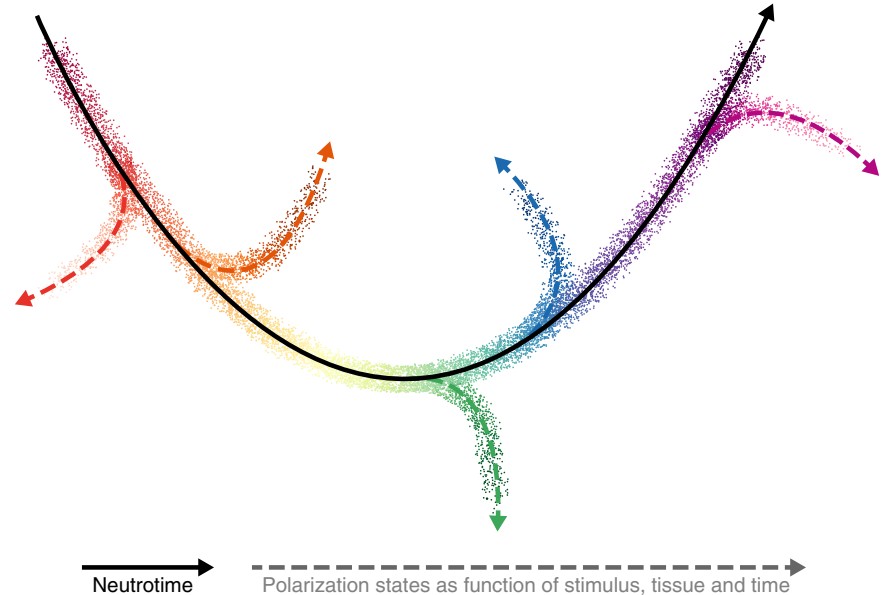

**Fig. 9 Proposed working model: neutrotime as the central organizing principle of neutrophil heterogeneity.** Healthy neutrophils are organized along one main sequence, termed neutrotime, from which they deviate as a function of time and environmental cues to reach different polarization states, orchestrated by shared and context-specific transcription factors. Experimental inflammation was found to recruit neutrophils predominantly near the mature pole of neutrotime; however, deviation from points earlier in the spectrum is also likely, reflected in arrows all along the neutrotime continuum. The colors of cells polarizing into different states illustrate that some features of the neutrotime sequence are maintained.

neutrophils recruited to sites of inflammation were generally far advanced along the neutrotime spectrum, suggesting that these cells were mature blood neutrophils before tissue entry. Many transcriptional changes were conserved across all disease conditions, likely reflecting a core transcriptional program engaged by neutrophils recruited into inflamed sites. This program includes upregulated genes for CD14, chemokines, and the IL-1 receptor antagonist. However, neutrophils from inflamed sites also exhibited considerable heterogeneity depending on experimental model. Neutrophils elicited over 3 h by instillation of IL-1β into lung and peritoneum were much more similar to each other than to those accumulating by day 7 of arthritis. Consistent with the enhanced lifespan of neutrophils in inflamed tissues, cells harvested from inflamed joints were further along the neutrotime axis, and they also exhibited upregulated *Cxcr4* transcript and CXCR4 protein, recently identified in zebrafish as a receptor helping to anchor neutrophils in inflamed tissues[47]. However, lung and peritoneal neutrophils remained distinguishable from each other, illustrating the importance of recruitment site to neutrophil phenotype. These findings are consistent with a recent survey of neutrophils across bone marrow, blood, spleen, lung, liver, skin and intestine under homeostatic conditions that found neutrophils entering different tissues to exhibit distinct signatures, promoted by tissue-specific factors such as CXCL12 in the lung[69].

Prior scRNA-seq studies of neutrophils have focused either on early development or within the context of tumor or inflammation[16,24,36]. By including mature neutrophils in blood and spleen, our data extend these results, identifying a continuous evolution of neutrophil gene expression with maturation that varies further with migration into inflamed sites. Early in neutrotime, gene expression is weighted toward anabolic functions and granule loading, while later in neutrotime effector genes such as *Il1b* become predominant. These findings echo early studies of human neutrophils sorted at distinct developmental stages[22]. Of note, a recent study of neutrophils in unperturbed tissues and with *E. coli* bacteremia, using graph-based cell clustering to partition neutrophils into discrete groups, suggested an

alternative model whereby neutrophils can sometimes bypass certain maturity stages[70]. Although our findings were otherwise highly concordant, we did not observe such discontinuities.

We thus propose a working model (Fig. 9) to understand heterogeneity among neutrophils. Neutrophils represent a single lineage organized along one main maturational sequence, neutrotime. As neutrophils progressing through this continuum encounter environmental cues, they deviate as a function of site, stimulus, and time, arriving at a phenotype that likely reflects both their starting position within neutrotime and the nature and sequence of signals encountered. For neutrophils recruited to inflamed sites in the models studied here, these cells primary emerge from the mature end of the spectrum, but the relative immaturity of circulating cells observed during inflamed states suggests that similar phenotypic divergence could occur earlier in neutrotime as well. This mechanism provides an opportunity to generate a highly diverse neutrophil repertoire without a requirement for committed developmental branches or cell subsets. Publicly available data for human neutrophils suggest a parallel pattern, though the full extent of congruity between human and mouse neutrophils remains to be determined.

Available through immgen.org via a dedicated visualization interface, the data presented here provide a public resource to help understand neutrophil biology. They establish that neutrophils in healthy mice represent a single developmental continuum, termed here neutrotime, characterized by continual evolution in gene expression from preNeus through mature circulating neutrophils. Further, they illustrate how this transcriptional evolution continues during recruitment to inflamed sites, varying with factors including site and stimulus. Together, these results establish a framework to understand the diverse neutrophil phenotypes observed under conditions of health and disease.

## Methods

**Mice, tissue preparation, and creation of single cell suspensions**. This study complied with ethical regulations for animal testing and research and was approved by the animal welfare committee of the Brigham and Women's Hospital (#2016N000535).

Male WT C57BL/6 J mice were obtained from The Jackson Laboratory (Stock No 000664) at age of 5 weeks and housed in SPF conditions for at least one week prior to the experiment, as per ImmGen protocol (https://www.immgen.org/Protocols/ImmGen%20Cell%20prep%20and%20sorting%20SOP.pdf). Housing conditions included temperature between 68 and 75 °F, humidity between 35 and 65% and lights on between 7 AM and 7 PM. Male and female K/BxN mice expressing the T-cell receptor transgene KRN and the major histocompatibility complex class II molecule Ag7 were housed in our animal facility at the Brigham and Women's Hospital and serum from male and female mice was obtained at age of 8–11 weeks and pooled. Experiments were approved by the animal care and use committees of the Brigham and Women's Hospital. Control mice were harvested at 6 weeks (dataset 1) and 8 weeks (dataset 2). Neutrophils were obtained from the circulation of healthy anesthetized mice by cardiac puncture: 1 ml of blood was collected by cardiac puncture in a syringe coated with EDTA (Invitrogen #15575-038, final concentration of EDTA: 5 mM). Blood, bone marrow and spleen were obtained from the same mice in each experiment. Following cervical dislocation, mice were immediately dissected to obtain the spleen and the tibias + femurs. The spleen was carefully cleaned of any attached fat and lymph nodes and then minced in a cell culture dish with the sterile back of a syringe to dissociate splenic immune cells. The dissociated splenic tissue was then passed through a 70-micron filter to create a single cell suspension. Bone marrow from femurs and tibia was flushed using 4 °C media to obtain bone marrow suspensions. All tissues were placed immediately into 4 °C media.

**K/BxN serum transfer arthritis.** Serum from 8 to 11-week-old K/BxN mice (both male and female) was pooled and injected intraperitoneally 150 μl into male WT mice aged 8 weeks on day 0 and day 2. K/BxN mice were obtained by crossing KRN TCR tg mice (background: C57BL/6) with NOD/Lt mice[71]. Arthritis was confirmed clinically and using a caliper to measure thickness of wrists and ankles. Mice were scored on day 6 (clinical score > 10 on a scale from 0 to 12) and were euthanized on day 7. Blood from arthritic mice was obtained by cardiac puncture. To obtain neutrophils from joints, we made a series of incisions on one side of the tibiotalar joint to allow drainage and then inserted a needle into the other side and flushed multiple times with cold PBS.

**Experimental pneumonitis.** Male WT mice aged 8 weeks were anesthetized with 100 mg/kg ketamine (Patterson Veterinary #07-892-5834) and 16 mg/kg xylazine (Patterson Veterinary #07-808-1947). 25 ng of recombinant Mouse IL-1β (R&D Systems #401-ML-005/CF) in 30 μL of phosphate-buffered saline (PBS, Corning #21-040-CV) were administrated intranasally. After 3 h, mice were euthanized and bronchoalveolar lavage (BAL) was performed using cold PBS as described[6].

**Experimental peritonitis.** Male WT mice aged 8 weeks were injected intraperitoneally with 25 ng IL-1β in 200 μL of PBS. After 3 h, mice were euthanized and peritoneal cells were harvested from the peritoneum with 5 mL of cold PBS as described[6].

**Neutrophil isolation.** All isolation solutions were at 4 °C to avoid activation of neutrophils. 4 °C phenol-red free DMEM (Sigma-Aldrich#D1145) + 0.1% sodium azide (Sigma-Aldrich #S2002-25G) + 10 mM HEPES (Gibco #15630-080) + 2% FCS (GeminiBio BenchMark #100-106) + 5 mM EDTA media was used in all cell manipulation steps. Single-cell suspensions were gently pelleted at 400 × g for 5 min at 4 °C and resuspended in 2 ml cold ACK Lysing Buffer (Lonza #10-548E) for erythrocyte lysis. After 3 min of lysis, medium was added and cells were pelleted at 400 × g for 5 min at 4 °C. Cells were then resuspended in cold media containing antibodies for staining.

**Flow cytometry analysis.** Single-cell suspensions from healthy peripheral blood, bronchoalveolar lavage, peritoneal lavage, arthritic joint and peripheral blood from arthritic mice were stained using LIVE/DEAD Fixable Blue Dead Cell Stain Kit (ThermoFisher #L23105) according to the protocol, with subsequent staining on ice with the respective antibody panel for 30 min (Supplementary Table 1). Neutrophils were gated based on FSC-A and SSC-A, doublets excluded in FSC-H vs. FSC-W and SSC-H vs. SSC-W, live cells selected and Ly6G-positive CD11b-positive neutrophils gated. Compensation was performed using single-stained compensation beads (ThermoFisher #A10513) and populations were selected based on fluorescence minus one (FMO) controls. All used antibodies, catalog numbers and dilutions and respective flow cytometry panels used are described in detail in Supplementary Table 1. Flow cytometry files were analyzed using BD FACSDiva version 8.0.1 and FlowJo version 10.6.1.

**Fluorescence activated cell sorting.** Prior to sequencing, single-cell suspensions were stained with anti-mouse Ly6G (clone 1A8)-Alexa Fluor 647 (BioLegend #127610) and anti-mouse CD11b (clone M1/70)-Brilliant Violet 421™ (BioLegend #101251) at 1:100 dilution, for 30 min on ice. 10 min before fluorescence-activated cell sorting, propidium iodide (Sigma-Aldrich #P4170) was added to a final concentration of 5 ng/ml. Neutrophils were gated based on FSC-A and SSC-A, doublets excluded in FSC-H vs. FSC-W and SSC-H vs. SSC-W, propidium iodide

negative live cells and finally Ly6G-positive CD11b-positive neutrophils were selected. Cutoffs for sorting were determined based on unstained, isotype and single-stained cells. Cells were sorted directly into PBS with a final concentration of 0.04% BSA (Millipore Sigma #A9647). The sorting strategy is shown in Supplementary Fig. 1a. All steps were performed on ice with cold reagents, and total time from mouse euthanasia to single cell encapsulation was <2 h.

**Droplet-based single-cell RNA-sequencing.** Sorted Ly6G-positive CD11b-positive neutrophils were loaded on a 10X Chromium device, following standard steps for library preparation, quality control, amplification and sequencing. Single Cell 3' v2 chemistry was used for datasets with healthy blood, bone marrow and spleen and v3 chemistry for experimental inflammation. Between 6000 and 12,000 cells were loaded per experiment and recovery of intact cells was between 20 and 47% depending on tissue. Average sequencing saturation for the datasets was 90.9% (range: 85.1–95.5%). Sequencing was performed on an Illumina HiSeq 4000 with 8 bp index read, 28 bp R1, and 96 bp R2 length reads. Reads were demultiplexed and aligned to the mm10 genome using Cell Ranger software (v1.2.0 for dataset 1, v2.1.0 for dataset 2 and v3.0.2 for dataset 3). The returned filtered cell barcode and feature matrices by CellRanger was then used for further analyses in R v3.6.1.

**Demultiplexing of hashtag oligo tagged multiplexed samples.** In order to minimize sample to sample variation, neutrophils sorted from K/BxN induced arthritis (blood and joint), IL-1β induced peritonitis (peritoneal neutrophils), IL-1β induced pneumonitis (alveolar neutrophils) and healthy control neutrophils from blood were tagged with hashtag oligonucleotides (TotalSeq™) directed against the abundantly-expressed CD45 and MHC I (diluted according to manufacturer instructions to 1.0 μg of antibody in 100 μl of staining buffer for every 1 million cells):

TotalSeq™-A0301 anti-mouse Hashtag 1 Antibody (BioLegend #155801, barcode ACCCACCAGTAAGAC, for K/BxN arthritis joint),

TotalSeq™-A0302 anti-mouse Hashtag 2 Antibody (BioLegend #155803, barcode GGTCGAGAGCATTCA, for K/BxN arthritis blood),

TotalSeq™-A0303 anti-mouse Hashtag 3 Antibody (BioLegend #155805, barcode CTTGCCGCATGTCAT, for IL-1β induced peritonitis),

TotalSeq™-A0304 anti-mouse Hashtag 4 Antibody (BioLegend #155807, barcode AAAGCATTCTTCACG, for IL-1β induced pneumonitis),

TotalSeq™-C0305 anti-mouse Hashtag 5 Antibody (BioLegend #155869, barcode CTTTGTCTTTGTGAG, for healthy blood).

Reads were assigned using Cite-Seq-Count (Roelli et al.[72]) package with the HTODemux function to assign HTOs. We detected barcodes from 5976 unique HTO tagged cells and 5537 distinct UMI barcodes and only retained the intersect of 5427 events for which both the HTO tag and the cell barcode had been detected. Only included HTO signals were then normalized using a centered log-ratio (CLR) transformation and demultiplexed in Seurat using k-means based clustering, a positive quantile cutoff of 0.999, a seed of 42. Cells were then assigned to one of the five clusters (n = 4856 cells), as negative (n = 17 cells) or as doublets (n = 554 cells). Doublets and HTO negative cells were excluded. 13,721 genes were detected in at least one of the remaining cells.

**Quality control and processing of single cell RNA-seq data.** We took the CellRanger filtered matrix of i rows representing genes and j columns representing barcodes (cells) as input. In the first 10X dataset, we detected 4078 cells overall: 1588 (38.9%) cells from blood, 1271 (31.2%) cells from bone marrow and 1219 (29.9%) cells from spleen. Counts with the same gene symbol and different transcript IDs were added. A total of 9112 genes were detected in at least one cell. In the second dataset, we recovered 12,829 cells overall: 4653 (36.3%) cells from blood, 3823 (28.8%) cells from bone marrow and 4353 (33.9%) cells from spleen. A total of 15,486 genes were detected in at least one cell.

As first step, we ran a cell classifier on the raw count matrix of all three 10X datasets as in Zemmour et al.[18]. We used the official ImmGen cell class expression set (Supplementary Data 2) containing gene expression data from 249 immune cell populations. We first divided the expression value for each gene within one population by the total expression of all genes within that cell population to obtain prior probabilities. We then filtered the expression matrix for genes that were detected in our respective datasets and employed a multinomial model to calculate the likelihood that each given single cell represents an a priori known ImmGen cell population. The resulting matrix contained cell barcodes as rows and known cell types as columns and known probabilities for each cell being of a specific known cell type. Each cell was then assigned to its most likely cell type. In the next step, cell types were summarized as higher lineages. All cells that did not correspond to any one of the six granulocyte datasets described in ImmGen (GN.Arth.BM, GN.Arth.SynF, GN.Bl, GN.BM, GN.Thio.PC, GN.UrAc.PC) were discarded. 3754 (92.1%) cells from the first dataset, 9225 (71.9%) cells from the second dataset and 4852 cells (99.9%) from the third dataset were retained for further analysis.

Next, we calculated the most likely cell cycle state for each cell using a curated list of characteristic transcripts for each cell cycle and an algorithm based on the pairs method that selects characteristic pairs of genes for each cell cycle phase whose relative expression difference is positive in the given cell cycle phase and negative in the other phases[23]. Cells were assigned to the S, G1, or G2M phase. For

downstream processing and normalization, we considered the difference between the S and G2M score of each cell and included it as co-variate to be modeled in the non-regularized linear regression model.

We then calculated summary statistics reflecting the quality of each cell, such as the number of detected genes and the number of unique molecular identifiers (UMIs) and the fraction of UMIs corresponding to mitochondrial features. 15,038 genes were non-zero in the combined three datasets containing 17,831 cells. Cells with more than 5% mitochondrial transcripts were removed (54 of 17,831) and cells with no available cell cycle information (5/17,777 cells) were removed. Cells with a number of unique features above the 99th percentile (1825 genes) were excluded as they represented possible doublets. Cells with a number of genes below the 1st percentile (280) were removed due to low quality. Overall, 3584 neutrophils from the first dataset, 9088 neutrophils from the second dataset and 4752 cells from the third dataset passed our rigorous quality control, retaining 17,424 cells and 15,038 genes. From cells retained in the analysis, the median number of detected genes was 515 and the median number of transcripts (UMIs) was 1455. Finally, only genes expressed in at least 10 cells were retained, so that the final expression matrix used for downstream analysis consisted of 10,900 genes × 17,424 cells. After Uniform Manifold Approximation and Projection (UMAP) clustering and inspection of defining gene expression (Supplementary Fig. 2), 53 debris events were excluded. The total number of reported cells in our final dataset is 17,371 cells. We found a strong relationship between the number of detected unique molecular identifiers (UMIs) and the number of detected transcripts (Supplementary Fig. 1l). There was also high agreement between the average expression of each gene in both datasets, with a Spearman correlation of 0.863 for blood, 0.878 for bone marrow and 0.849 for spleen (Supplementary Fig. 1k).

**Data alignment**. We used regularized negative binomial regression and a recently-described canonical correlation-based integration workflow, implemented in Seurat, to obtain a combined expression matrix for the first two datasets encompassing healthy cells[73]. First, we calculated highly variable genes from both datasets using a residual variance cutoff greater than 1.3, modeling each gene after a negative binomial regression model encompassing the number of detected UMIs, the difference between the S and G2M cell cycle score and the percentage of mitochondrial UMIs as covariates. All highly variable genes from both datasets were used for integration. We then integrated the two datasets using SCTransform with L2 normalization on the CCA cell embeddings after dimensional reduction, again using the same negative binomial regression parameters. 3322 robustly expressed genes shared between the datasets of healthy cells were retained for downstream analysis. For cells obtained from experimental inflammation, no integration had to be performed, as they were all captured on the same 10X run, and we therefore only applied normalization as detailed above.

**Dimensionality reduction by principal component analysis and UMAP**. We used the integrated dataset of all healthy cells to run a combined principal component analysis (PCA), computing 50 principal components, of which the first 20 (based on an elbow plot) were used to compute a Uniform Manifold Approximation and Projection (UMAP) embedding of the cells. Next, we clustered cells, exploring a range of different resolution settings of $k = 20/100/500$ and resolution of 0.3/0.8/1.5. Following visual inspection of UMAP plots and resulting heatmaps with cells separated by cluster identity, we found that a small population of 53 cells were debris.

**Diffusion maps**. We used the same twenty principal components as above used to calculate the UMAP embedding of cells to compute a cell to cell distance matrix, where the difference between cells was calculated as (1 − Pearson correlation). We used this distance matrix to compute a diffusion map with standard parameters and density normalization and rotate enabled. We manually chose the preNeu cells as root cells and extracted pseudotime values along the resulting trajectory. We scaled and centered the diffusion components 1 and 2 and scaled the pseudotime values between 0 and 1.

**RNA velocity analysis**. We processed the fastq files for each experiment into loom files for each sample using the velocyto v0.17 package[32]. We then took the resulting loom file and processed it into both a spliced and unspliced gene by cell matrix. Using the velocyto R package, we took the most variable genes in each experiment (found using Seurat) and calculated gene relative velocity RNA estimates using default settings and a cell nearest neighbor value of 300. These gene estimates were then visualized on our 2D diffusion map embedding.

**Validation of key neutrotime transcripts in Hoxa9 RNA-Seq data**. We obtained a count matrix of RNA-seq data of developing Hoxa9 cells at different time points after estrogen withdrawal. We used the 25 genes with strongest positive and strongest negative Spearman correlation with neutrotime and examined their expression profiles at the time points of 96 and 120 h (representing the last maturation phase of developing neutrophils). We then computed the average fold change from 96 to 120 h of these top neutrotime-associated genes in $\log(x + 1)$ transformed expression values.

**Obtaining a preNeu gene score for resting neutrophils**. We obtained a raw RNA-seq count matrix from Evrard et al. (GSE109467) and summed gene expression for genes with the same symbol and multiple transcript IDs. We used limma to model each gene as a linear model with samples assigned as either preNeu or all other groups. We found 386 transcripts with a Benjamini–Hochberg corrected (corresponding to the false discovery rate) $p$ value ≤ 0.05 and an absolute $\log_2$-fold-change ≥ 1.5 that robustly defined preNeus compared to other neutrophil developing stages.

**Gene expression along neutrotime**. We used a matrix containing normalized gene expression from all healthy cells and calculated the spearman correlation coefficient for all 3322 genes with neutrotime. For visualization, we chose the 50 genes with highest and lowest correlation with neutrotime, respectively. For the heatmap, we scaled the expression of each gene between 0 and 1. We then visualized smoothed gene expression along neutrotime for select genes with an early, intermediate or late peak using generalized additive models with integrated smoothness estimation (standard settings).

**Gene clustering**. We used a gene × cell expression matrix of healthy neutrophils from all tissues from the first two datasets and ordered the cells according to their neutrotime score. We then performed $k$-means clustering on this matrix with ten iterations and $k = 10$, yielding nine informative clusters and one cluster with 6 remaining heterogeneous genes.

**Interferon response in neutrophils**. To test the association of neutrophils and interferons, we first obtained gene expression data from 10 ImmGen cell types (neutrophils, gamma-delta T cells, CD4 T cells, CD8 T cells, B cells, dendritic cells, macrophages, NKT cells, NK cells and regulatory T cells) isolated from the spleen of mice 2 h after subcutaneous injection of 10,000 IU IFN α (Supplementary Data 3)[35]. We assigned each gene to a cell type which displayed the highest $\log_2$ fold change after IFN stimulation. We ordered the resulting matrix by highest expression and quantitated the number of highest fold change of genes for each cell type. We obtained gene sets characterizing Type I interferon response (type I interferon signaling pathway; GO:0060337) and type II interferon response genes (interferon-gamma-mediated signaling pathway; GO:0060333). We examined the expression of these genes along neutrotime. As neutrophils displayed the highest gene set changing after IFN alpha stimulation, in line with the strong variance of many type I interferon response genes along neutrotime, we performed principal component analysis followed by UMAP dimensionality reduction based exclusively on genes included in GO:0060333. Cells were then colorized by source tissue, neutrotime and select Type I interferon response genes.

**Mapping of datasets onto neutrotime**. We selected genes with a Moran's I score ≥ 0.2 and a mean expression ≥ 0.1 and categorized them into three classes based on their spearman correlation $R$ with neutrotime: 1. Positively correlated genes ($R \geq 0.2$; 29 genes; group of genes represented as $N_+$) 2. Negatively correlated genes ($R \leq -0.2$; 30 genes; group of genes represented as $N_-$), 3. Stably expressed genes ($-0.1 \leq R \leq 0.1$; 141 genes; group of genes represented as $N_o$).

We recalculated neutrotime directly from gene expression space, where $e(g)$ represents the expression value $e$ of a gene $g$:

$$neutrotime_S\ score = \frac{\sum_{g \in N_+} e(g) - \sum_{g \in N_-} e(g)}{\sum_{g \in N_0} e(g)} \tag{1}$$

For analysis of single cell transcriptomic data obtained in this manuscript, log-transformed, normalized counts were used. For visualization in Fig. 5a, only genes with mean expression ≥ 0.1 (same threshold as above) were shown. We plotted neutrotime and neutrotime-S against each other and again calculated the Spearman correlation between the two scores for each individual cell. We performed the same analysis on RNA-seq data obtained from Evrard et al. (GSE109467) using raw counts. For statistical testing of neutrophils from inflammation, we first performed an ANOVA followed by Dunnett's test for comparing inflamed groups with the blood control.

**Assessing neutrotime in human neutrophils**. We obtained human single-cell RNA-Seq data from 40,000 bone marrow cells from the human cell atlas[38], available through the SeuratData package[39]. We next integrated the bone marrow expression data from eight different donors using the same approach illustrated above under "Data alignment". Next, we used genes with Spearman correlation with neutrotime ≤ 0.25 as early neutrotime genes and ≥ 0.25 as late neutrotime genes and mapped them to human 1:1 orthologs with high confidence using ENSEMBL version 100 (Supplementary Data 4). Next, we calculated a module score for early and late neutrotime as described by Tirosh et al.[74]

**Gene set enrichment analysis and GO term analysis along neutrotime**. GSEA was performed using GSEA v4.0.3 (https://www.gsea-msigdb.org/gsea/index.jsp)[75,76]. We used a list of genes based on their Spearman correlation with neutrotime as a pre-ranked list to run GSEA. All 8 major collections from the Molecular Signature Database v7.1 were used. The dataset was collapsed to symbols

using the corresponding Chip platform from the Broad Institute (ftp.broad-institute.org://pub/gsea/annotations_versioned/Mouse_Gene_Symbol_Re-mapping_to_Human_Orthologs_MSigDB.v7.1.chip) R was used to create a ranked list of gene sets that was plotted against false discovery rate (Fig. 3d), alongside with the normalized enrichment score (Fig. 3e). Significantly enriched genes were sorted by the absolute value of the normalized enrichment score and colored by the false discovery rate (Fig. 3f).

Gene ontology enrichment analysis was performed and visualized using the R package topGO v2.38.1[77] (Supplementary Fig. 5). The corresponding GO term accessions were extracted from ENSEMBL v98, genome assembly GRCm38.p6 using the R package BiomaRt v2.42.1[78,79]. The Spearman correlation table was separated and filtered into a positive (Spearman's $R > 0.2$) and a negative (Spearman's $R < -0.2$) list. Gene to GO term mappings were generated as described by the authors, enrichment tests were performed using the arguments algorithm='classic', statistic='fisher'.

**Spatial autocorrelation of genes along neutrotime**. Cells and genes were filtered as described above. We then generated and pre-processed a scRNA-seq dataset using the R package Monocle3 v0.2.1 as described[25,80,81]. After pre-processing, batch correction was performed with fast mutual nearest neighbors correction[82] and using the R package batchelor v1.2.4 as described by the authors with few modifications (cos.norm = FALSE, pc.input = TRUE, k = 5). We calculated the first 50 principal components and performed dimensional reduction to 2-dimensional space using UMAP[83]. Cells were colored by tissue and plotted (Supplementary Fig. 4). Unsupervised clustering was performed using community detection with the Leiden algorithm (Levine et al.[84] and Traag et al.[85]) embedded in monocle. The principal graph was calculated as described by the authors of the Monocle3 package (Supplementary Fig. 4). We ordered the cells by manually selecting bone marrow as the start node in the principal graph to color the cells by pseudotime (Supplementary Fig. 4). In addition, we colored the cells using the same four clusters as obtained from the UMAP embedding presented in the main manuscript.

To find genes that vary between groups of cells in UMAP, we applied Moran's I, a measure of spatial autocorrelation, which has recently been described to be effective in single-cell RNA-seq[81]. Moran's I was calculated as described by the authors of Monocle3[25]. The k-nearest neighbor graph was used as input. The results were then plotted against the genes' correlations with neutrotime, with size representing mean expression level (Fig. 5a).

**Inferring transcription factor activity**. Transcription factors associated with early- and late-stage neutrotime were predicted using ChIP-X Enrichment Analysis Version 3 (ChEA3) as described by the authors[40]. Briefly, gene lists are compared to 6 annotated TF target libraries including human, mouse and rat data by a Fisher's Exact Test with a background size of 20,000. We used the MeanRank output as the ChEA3-score for downstream analysis. As gene lists, we submitted the top 100 genes with the highest Spearman correlation with neutrotime as a representation for late-stage neutrotime and the top 100 genes with the lowest Spearman correlation with neutrotime as a representation for early-stage neu-trotime, respectively. The results were robust for variations of gene list lengths (Supplementary Fig. 7). In addition, we extracted cell trajectories from the IL-1β- and the K/BxN serum-induced inflammation models, ordered the cells along the trajectories and extracted the top 100 genes with the highest (most advanced along the trajectory) and lowest (least advanced in the trajectory) correlation to analyze the transcription factor activity in a similar way (Fig. 8b–d).

**Transcription factor expression along neutrotime**. To explore TF expression along neutrotime, we filtered the expression matrix using a curated list of mouse TFs (Riken Mouse Transcription Factor Database). We scaled each gene between 0 and 1 for heatmap visualization and ordered all cells along neutrotime. We plotted smoothed expression of select TFs along cells ordered along neutrotime. To explore inferred TF regulatory activity in early and late neutrotime, we log-transformed the multiplicative inverse of each gene's ChEA3 score for early and late neutrotime and plotted them against each other. In this analysis, a high value indicates high reg-ulatory activity and low value indicates a low regulatory activity. We flagged the overlap of the 50 TFs that displayed the greatest regulatory activity as well as the 50 TFs with greatest activity difference in early neutrotime ($n = 18$) or in late neu-trotime ($n = 23$) and obtained the unique ($n = 30$) TFs from this analysis for plotting. A similar analysis was performed using TF activity in cells ordered along the IL-1 and K/BxN trajectories. We then plotted a binary matrix of regulatory activity from genes included in the test set as well as their putative TFs to highlight the diverse TFs driving the gene expression signature along neutrotime. Finally, we analyzed the inferred TF activity of TFs (ChEA3 score) versus actual TF expression (Spearman correlation with neutrotime) to identify TFs that are both highly active as well as expressed on the transcript level.

**Computing shared and distinct inflammatory response programs in neu-trophils**. We performed a Wilcoxon rank-sum test on gene expression across all ten pairwise comparisons of the experimental inflammation data. P values were corrected using Bonferroni correction with all genes in the dataset. To obtain genes

with a conserved response in inflammation, we selected genes with average fold change ≥ 0.5 in the conditions to be tested compared to the control conditions. For heatmap visualization we calculated the average of all cells of each condition.

**R analysis packages**. R v3.6.1 was used for downstream analysis of single cell RNA-Sequencing data.

The following packages were used in analysis: batchelor (v1.0.1), biomaRt (v2.42.1), Biostrings (v2.54.0), ChEA3 (v3), data.table (v1.12.9), DescTools (0.99.34), destiny (v3.0.1), diptest (v0.75-7), edgeR (v3.28.0), egg (v0.4.5), ggplot2 (v3.3.0), ggrepel (v0.8.2), gridExtra (v2.3), GSEA (v4.0.3), h5 (v0.9.9), leiden (v0.3.1), limma (v3.42.0), MASS (v7.3-51.5), Matrix (v1.2-18), matrixStats (v0.56.0), Monocle3 (0.2.1), org.Mm.eg.db (v3.10.0), pheatmap (v1.0.12), princurve (v2.1.4), RColorBrewer (v1.1-2), readxl (v1.3.1), reshape2 (v1.4.4), scales (v1.1.0), scater (v1.14.6), scran (v1.14.5), Seurat (v3.1.0), tidyverse (v1.3.0), topGO (v2.38.1), velocyto.R (v0.6), viridis (v0.5.1).

Flow cytometry files were analyzed using BD FACSDiva version 8.0.1 and FlowJo version 10.6.1.

**Statistics and reproducibility**. Unless otherwise indicated, $N = 3$ mice were pooled per tissue and per experiment. scRNA-seq studies were performed two times for healthy tissues (shown in Figs. 1–6 and Supplementary Figs. 3, 5–9) and once for neutrophils in experimental inflammation (shown in Figs. 7–8 and Supplementary Fig. 10). Flow cytometry experiments for Supplementary Fig. 11 were performed with 4–5 biological replicates depending on condition; the average MFI change relative to healthy blood is shown. Details for the used statistical tests follow: Figure 1a, b, d, e, f, g, h, i: descriptive analysis. Figure 1c: Marker genes identified by Wilcoxon Rank Sum test (two-tailed) using the Seurat function "FindAllMarkers" with standard settings; only genes with $\log_e$ fold change ≥ 0.25 and Bonferroni adjusted $p$-value ≤ 0.05 are shown. Figure 1j: Unpaired $t$-test (two-tailed) between preNeu and all other neutrophils within each tissue. Figure 1k: ANOVA followed by unpaired $t$-test (two-tailed) for all three pairwise comparisons between tissues (all comparisons not significant).

Figure 2: Descriptive analysis.

Figure 3a–c: Descriptive analysis. Figure 3d–f: Gene Set Enrichment Analysis as previously described.[76] In short, a normalized enrichment score was calculated by going through a list of genes based on their Spearman correlation with neutrotime as a pre-ranked list and calculating a running-sum statistic, which was then normalized for differences in the sizes of the gene sets that were looked at (normalized enrichment score). To account for multiple hypothesis testing, an FDR approach was used to maintain a defined level of significance, values < 0.01 were considered. Figure 4: Descriptive analysis. Figure 5a: Descriptive analysis. Figure 5b: R indicates the Spearman's rank correlation coefficient. Figure 5c: One-way ANOVA ($P < 0.0001$) followed by Tukey's multiple comparison test (two-tailed). Figure 5d: Differential expression analysis. Limma-voom was used to fit a linear model, calculate empirical Bayes statistics and derive differential expression. Genes with a $\log_2$ fold change threshold of |1| and a Benjamini & Hochberg adjusted $p$-value (corresponding to false discovery rate) threshold of ≤ 0.05 were considered. Figure 5e–g: descriptive analysis. Figure 6: descriptive analysis. Figure 7a: schematic. Figure 7b, e: descriptive analysis. Figure 7c: ANOVA followed by Dunnett's multiple comparison test (two-tailed) compared to healthy blood. Figure 7d: As for Fig. 1c, marker genes were identified by Wilcoxon Rank Sum test (two-tailed) using the Seurat function "FindAllMarkers" with standard settings; only genes with log fold change ≥ 0.5 compared to healthy blood and Bonferroni adjusted $p$-value ≤ 0.05 were considered. For comparisons that examined changes between multiple groups compared to healthy blood (e.g. "Inflamed tissues"), as most conservative approach, the highest adjusted $p$-value was chosen for each gene. Figure 7f: ANOVA followed by Dunnett's multiple comparison test (two-tailed). Percentage of cells with non-zero expression shown as descriptive statistic.

Figure 8a: As in Fig. 1c, marker genes for each condition were identified by Wilcoxon Rank Sum test (two-tailed) using the Seurat function "FindAllMarkers" with standard settings; only genes with log fold change between conditions ≥ 0.25 and Bonferroni adjusted $p$-value ≤ 0.05 were considered. The list of marker genes was subsetted to transcription factors to display differentially expressed transcription factors between conditions. Cells were randomly downsampled to 200 cells per condition for plotting only.

Figure 9: Schematic representation.

**Reporting summary**. Further information on research design is available in the Nature Research Reporting Summary linked to this article.

## Data availability

Single cell RNA sequencing data have been deposited in the Gene Expression Omnibus (GEO) database under the accession code GSE165276. Interactive data browsing is available via the ImmGen single cell explorer. The following publicly available datasets were used in this study: Riken Mouse Transcription Factor Database (TFdb; http://genome.gsc.riken.jp/TFdb/) GO terms "Type I interferon signaling pathway" (https://www.ebi.ac.uk/QuickGO/term/GO:0060337) "Interferon-gamma-mediated signaling pathway" (https://www.ebi.ac.uk/QuickGO/term/GO:0060333) Evrard et al. bulk RNA-Seq data (GSE109467) Sykes et al. bulk RNA-Seq data of developing Hoxa9 cells

(GSE84874) ImmGen class means of 259 unique cell populations (included in this manuscript as Supplementary Data 2) ImmGen in vivo IFN gene expression change at 2 h (included in this manuscript as Supplementary Data 3). Source data are provided with this paper. All other data are provided in the article and its Supplementary files or from the corresponding author upon reasonable request.

## Code availability

No custom software was written for this manuscript. Methods describe the used analytical tools and variables. R code is available from the corresponding author upon request.

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

## Acknowledgements

The Immunological Genome Project is funded by NIH/NIAID award R24AI072073. R.G-B. was funded by an MD fellowship from Boehringer Ingelheim Fonds, a physician scientist development grant from the Medical Faculty Heidelberg and a research grant from the German Society for Rheumatology (DGRh). F.A.R. is funded by an MD fellowship from Boehringer Ingelheim Fonds and two Joint Biology Consortium microgrants from parent award P30AR070253. P.C. is funded by the Arthritis National Research Foundation and the Gilead Sciences Research Scholars Program in Rheumatology. P.A.N is supported by NIH/NIAMS awards R01AR065538, R01AR075906, R01AR073201, R21AR076630, P30AR070253, R56AR065538, NIH/NHLBI R21HL150575, a Lupus Research Alliance Target Identification in Lupus Grant, the Fundación Bechara, and the Arbuckle Family Fund for Arthritis Research. Some illustrations were created with BioRender.com.

## Author contributions

R.G-B. selected the inflammatory models to be used in the study, acquired neutrophils from healthy and inflamed mice, processed cells for single cell analysis, performed computational analysis, conceptualized *neutrotime* and wrote the manuscript. F.A.R. provided important conceptual input in experimental planning, performed K/BxN serum injections, induced IL-1β peritonitis, IL-1β pneumonitis, harvested healthy and inflamed neutrophils, performed computational analysis and wrote the manuscript. P.C. provided important conceptual input in experimental planning, performed K/BxN serum injections, induced IL-1β peritonitis, IL-1β pneumonitis and harvested healthy and inflamed neutrophils. G.S. acquired neutrophils from healthy mice, processed cells for single cell analysis and performed computational analysis. A.L. provided important conceptual input in experimental planning, performed K/BxN serum injections and harvested healthy and inflamed neutrophils. B.V. provided important conceptual input in experimental planning and data analysis and prepared data display on the ImmGen website. N.N-M. performed K/BxN serum injections and harvested healthy and inflamed neutrophils. R.B. performed K/BxN serum injections and harvested healthy and inflamed neutrophils. P.A.M. conceived the project, provided important conceptual input in data analysis, interpretation and validation and supervised the entire project. P.A.N. conceived the project, contributed the infrastructure for experimental work, guided data analysis, interpretation and validation, supervised the entire project and wrote the manuscript. ImmGen Consortium. The Immunological Genome Project Consortium established the framework to realize this study, contributed resources, assisted in experimental design, and contributed to data analysis and editing of the manuscript.

## Competing interests

The authors declare no competing interests.

## Additional information

## ImmGen Consortium

Oscar Aguilar[6], Rhys Allan[7], Jilian Astarita[8], K. Frank Austen[1], Nora Barrett[1], Alev Baysoy[9], Christophe Benoist[9], Brian D. Brown[10], Matthew Buechler[8], Jason Buenrostro[11], Maria Acebes Casanova[12], Kaitavjeet Chowdhary[9], Marco Colonna[13], Ty Crowl[14], Tianda Deng[14], Fiona Desland[12], Maxime Dhainaut[10], Jiarui Ding[15], Claudia Dominguez[8], Daniel Dwyer[1], Michela Frascoli[15], Shani Gal-Oz[16], Ananda Goldrath[14], Ricardo Grieshaber-Bouyer[1,2], Tim Johanson[7], Stefan Jordan[12], Joonsoo Kang[15], Varun Kapoor[8], Ephraim Kenigsberg[12], Joel Kim[12], Ki wook Kim[13], Evgeny Kiner[9], Mitchell Kronenberg[16], Lewis Lanier[6], Catherine Laplace[9], Caleb Lareau[11], Andrew Leader[12], Jisu Lee[17], Assaf Magen[12], Barbara Maier[12], Alexandra Maslova[18], Diane Mathis[9], Adelle McFarland[13], Miriam Merad[12], Etienne Meunier[18], Paul A. Monach[1], Sara Mostafavi[18], Soren Muller[8], Christoph Muus[19], Hadas Ner-Gaon[17], Quyhn Nguyen[14], Peter A. Nigrovic [1,5,21✉], German Novakovsky[18], Stephen Nutt[7], Kayla Omilusik[14], Adriana Ortiz-Lopez[9], Mallory Paynich[16], Vincent Peng[13], Marc Potempa[6], Rachana Pradhan[8], Sara Quon[14], Ricardo Ramirez[9], Deepshika Ramanan[9], Gwendalyn Randolph[13], Aviv Regev[19], Samuel A. Rose[10], Kumba Seddu[9], Tal Shay[17], Avishai Shemesh[6], Justin Shyer[8], Christopher Smilie[19], Nick Spidale[15], Ayshwarya Subramanian[19], Katelyn Sylvia[15], Julie Tellier[7], Shannon Turley[8], Brinda Vijaykumar[9], Amy Wagers[11], Chendi Wang[18], Peter L. Wang[13], Aleksandra Wroblewska[10], Liang Yang[9], Aldrin Yim[13] & Hideyuki Yoshida[20]

[6]Department of Microbiology & Immunology, University of California San Francisco, San Francisco, CA, USA. [7]The Walter and Eliza Hall Institute of Medical Research, Parkville, VIC, Australia. [8]Department of Cancer Immunology, Genentech, South San Francisco, CA, USA. [9]Department of Immunology, Harvard Medical School, Boston, MA, USA. [10]Icahn School of Medicine at Mount Sinai, New York, NY, USA. [11]Department of Stem Cell and Regenerative Biology, Harvard University, Cambridge, MA, USA. [12]Immunology Institute and Tisch Cancer Institute, Icahn School of Medicine at Mount Sinai, New York, NY, USA. [13]Department of Pathology and Immunology, Washington University School of Medicine, St. Louis, MO, USA. [14]Division of Biological Sciences, University of California San Diego, La Jolla, CA, USA. [15]Department of Pathology, University of Massachusetts Medical School, Worcester, MA, USA. [16]La Jolla Institute for Immunology, La Jolla, CA, USA. [17]Department of Life Sciences, Ben-Gurion University of the Negev, Be'er Sheva, Israel. [18]Department of Statistics, University of British Columbia, Vancouver, BC, Canada. [19]Broad Institute of Massachusetts Institute of Technology and Harvard, Cambridge, MA, USA. [20]YCI Laboratory for Immunological Transcriptomics, RIKEN Center for Integrative Medical Sciences, Kanagawa, Japan.

