## [Peer Review File · Nature Communications]

REVIEWER COMMENTS

Reviewer #1 (neutrophils, cytokines, chemokines) (Remarks to the Author):

In this study, Grieshaber-Bouyer and colleagues applied single-cell RNA sequencing to murine neutrophils isolated from different biological tissues. The main result of the study is the identification of a neutrophil maturation trajectory, defined neutrotime. The study and all the bioinformatic analyses are accurate and well done. The major issues are related to the absence of comparative analysis in human neutrophils precursors and the lack of analysis of precursors, prior to PreNeu step, which would have allowed to validate the neutrotime even better.

Major points

1. On page 5, the authors point out that a broad correspondence between Monocle's pseudotime analysis and the clusters identified by UMAP is recognizable. This is true for P1 and probably for P4, but not for P2 and P3. In fact the P2 and P3 clusters identified by UMAP were composed mainly of BM cells while using Monocle P2 seems composed principally of spleen cells (extended Data Fig4a). The spleen is a hematopoietic organ in mice and one should expect the presence of a large number of neutrophil precursors, and therefore more in agreement with Monocle's analysis. Moreover, gating strategy to isolate neutrophils, as well as Ly6G and CD11b membrane expression from all three different sources (BM, blood and spleen), should be shown.

2 The indication that neutrophils exhibit an interferon signature is not evident from Figure 4. The authors should carefully revise the figure because there is no correspondence with the text. Only IFITM genes are differentially regulated along neutrotime and moreover not all in the same direction (Ifitm2 and Ifitm1 increased progressively with neutrotime while Ifitm6 was expressed preferentially early in neutrotime). All the other ISGs are not modulated therefore authors cannot refer to an interferon signature.

3 In a recent study (PMID: 32579887) the group of Ng has identified two neutrophils precursors more immature than preNeu (proNeu1 and proNeu2). Authors should perform single-cell RNAseq also in this newly discovered cell types to fully validate their neutrotime model.

4 Why Itf (lactoferrin), a specific granule protein, is included in the Combined ChEA3 and TF expression analysis of Figure 6e which should include only transcription factors. Authors should carefully check the list of transcription factors analyzed

Minor points

1. HTOs were used also for datasets 1 and 2? Or different 10X runs were performed for blood, bone marrow and spleen. If this last is the case, how many cells were loaded per run?

2. Using bulk RNA-seq Evrad found that in the bone marrow are present mature neutrophils with similar characteristics to blood mature neutrophils. Instead, in the present study, using scRNA-seq authors find major differences (UMAP of figure 1d) and this should be at least discussed.

3 Which are the differentially expressed genes among neutrophils from blood or spleen? Even if minimal, is there an effect on gene expression of tissue environment?

4 In dot plots of fig1d, fig 4e (upper panel) and extended Data Fig4a (left panel) dots are too big and mask dots positioned below. For example, in extended Data Fig4a it seems that very few blood cells have been analyzed

5 Page 6: Extended Data Fig4d is mentioned in the text before Extended Data Fig4b

6 Fig. 3d-f: Authors state that: "Gene Set Enrichment Analysis (GSEA) similarly indicated enrichment of granule loading with early neutrotime and effector processes with late neutrotime (Fig. 3d-f)". However, in the figure, the indication of the enrichment of granule loading should be better evidenced because absolutely not clear. Moreover, the most enriched gene sets are related to lymphocyte and not granule, this should be discussed

7 misspelling on page 13 last row: "are" instead of "ate"

Reviewer #2 (granulocytes, neutrophils) (Remarks to the Author):

This paper identifies a chronologic developmental spectrum of neutrophil phenotypes, which the authors call neutrotime. Their data provide a strong argument for the notion that the various neutrophil subsets which can be identified actually reflect a single developmental continuum.

The study is well thought out, well executed, and very innovative. This being said, there remains a few discussion aspects that should be addressed (listed below), which would widen the scope of the findings.

1. The data focuses on gene expression patterns in various neutrophil subsets, but to which extent do these patterns relate to surface marker patterns observed in similar subsets, and in other well characterized neutrophil subsets (e.g. N1, N2), in vivo? There exist enough such data in the literature to start drawing parallels in the Discussion.

2. Though the concept of neutrotime is likely to apply to human neutrophils as well, the authors should here again compare their data with patterns observed in corresponding or similar human neutrophil subsets. Because there exist so many striking differences between human and mouse neutrophils, where would one expect to see divergent patterns between the two species? This is obviously where this field of study is leading, so some discussion is warranted.

Reviewer #3 (multiparameter flow, neutrophils) (Remarks to the Author):

In this paper, the Nigrovic group sets out to address an important and timely issue that is presently under intensive investigation in the "neutrophil community". Neutrophil heterogeneity is receiving substantial attention and it remains to be fully established whether this heterogeneity stems from distinct ontogenetic developmental paths that are defined by distinct molecular programs or whether heterogeneity stems from plasticity of the respective neutrophil differentiation stages in response to environmental cues.

In this paper, the authors present a thorough single cell transcriptomic analysis of murine neutrophils under homeostatic conditions. They nicely outline the description of a differentiation program, that they decided to name "neutrotime". In agreement with the authors, this reviewer agrees that the provided data sets and the data analysis represent a valuable resource for the research community. In the opinion of this reviewer, the connection and transition of homeostatic neutrotime to situations of inflammation, is less well-described in this otherwise excellent paper.

Conceptual critique:

Authors nicely outline in figure 1-6 the neutrotime transcriptional differentiation program in mice under non-inflamed homeostatic conditions. To this end, they investigate the neutrotime in blood, spleen and bone-marrow. In figure 7-8 the authors investigate neutrophils from blood, joint, lung and peritoneum of "arthritic" mice and mice exposed to IL-1b-driven inflammation.

Grieshaber-Bouyer et al. claim (in the abstract and other parts of the paper) that experimental inflammation translates into structured permutation of the neutrotime program. In the final part of the results section as well as in the final part of the discussion, the authors state and/or imply that during inflammation neutrophils follow the established neutrotime continuum. Variability under inflammatory conditions should then stem from the varying extrinsic inflammatory factors that modulate mature neutrophils that reached this state through a common neutrotime program. With reference to the last sentence of the results section (page 12), a key question comes up: Do the authors show in this paper that the neutrotime continuum / programe is not perturbed and rather remains stable in the presence of inflammation, and in particular IL-1-driven inflammation?

When first reading the abstract, I expected to see an analysis of neutrotime, executed on bone-marrow, blood and spleen neutrophils under inflammatory conditions (in addition to naïve mice). Is

neutrotime perturbed under inflammatory conditions? Why do the authors fail to show a direct comparison of neutrotime in naïve and inflamed mice.
Except for figure 7c, Fig 1-6 neutrotime/healthy is not well connected to Fig 7-8 inflammation.

Major critique:

1. Figure 2d shows that the vast majority of neutrophils in blood, BM and spleen belongs to two subpopulations that are located just before 0.25 and at the very end of the neutrotime scale. For the reader and people working in the field, it would be important to know, how this analyses would look like under conditions of inflammation.
2. Ext data figure 4c: It appears that neutrotime is not executed at "constant pace". Rather two rapid maturation steps are included. Both are associated with the maturation of the two major subsets shown in figure 2d. This step-wise differentiation and maturation is not really emphasized in the text. Do I misinterpret the data or is this interpretation correct. If not correct, please explain in the rebuttal letter why not correct. If correct, shouldn't this be emphasized more in the manuscript?
3. Ext data figure 6: The transcriptional changes occurring in the 0.0 to 0.5 (first half) of neutrotime are nicely reflected in this figure. However, there is no cluster that is associated with the final steps of neutrophil maturation in the latest phase of neutrotime. Why is this so?
4. Ext data fig 6 – 8; and figure 2-4: Similar papers in the field combine RNASeq data and protein expression data. In this paper, we exclusively look at transcriptomic data. Is there a way that authors could translate some findings to the protein level? For example, transcriptomic data for chemokine receptors is important to show. However, surface expression of chemokine receptors is largely regulated by exposure to their ligands – including ligand-induced receptor downregulation. Second example; granule protein expression is only partly regulated on the transcriptomic level. Neutrophil activation and degranulation greatly influence the protein level of those components. In other words: Are we looking here at transcriptional profiles or are we also looking at biologically active protein profiles?

Minor issues:

1. Ext data Fig 1: what are the Ly6G negative cells in panel B? According to panel A, only Ly6G positive cells are within the gate?
2. Expression of Ly-6G and CD11b differs between blood, spleen and BM neutrophils (and lung/peritoneum/joint neutrophils) and between naïve and inflamed mice. As these gates and FACS are the starting point for all analyses, it would be nice to extend the "ext data fig 1a" accordingly and show all starting populations.
3. Color code of P1-4 does not seem to match the populations shown in UMAP plots.

Reviewer #4 (single cell analysis, RNAseq) (Remarks to the Author):

The manuscript by Grieshaber-Bouyer et al aims to study neutrophils subsets in health and disease. Understanding Neutrophil heterogeneity and how different neutrophil subsets are functionally distinct is a timely and important question. To achieve this, the authors use scRNA-seq to decipher cell subsets based on their transcriptional profiles. Importantly, this method requires elaborative analysis to achieve its potential, as well experimental follow up to validate the reliability of the findings. Unfortunately, the data presented in this paper, of 4 main subsets, fails to expand our knowledge as these exact findings were already reported (DOI:<https://doi.org/10.1016/j.immuni.2018.02.002>). Furthermore, the data fails also to discriminate between neutrophil subsets in different tissues in health and disease (e.g., bone marrow blood and so on). Last, the authors do not provide functional experiments to support their scRNA-seq analysis.

Major comments:

1. Using scRNA-seq, the authors aim to understand neutrophil subsets within different organs. The authors chose the 4-cluster model (fig. 1b) as their main classifier of neutrophil subsets in the paper. However, similar subsets that define developmental trajectory and functional properties of neutrophils in bone marrow, blood and tissues was already studied (see the DOI above). Thus, the

finding of 4 clusters is not novel. It is surprising that the authors did not look for possible differences in models with higher cluster number (sup figure 2a) or further apply knn on each cluster to look for delicate changes between subpopulations in each cluster. The scRNA-seq analysis, which is the focus of the paper, is not comprehensive.

2. Even more confusing is the scRNA-seq analysis of the inflammation models. First, the choice of different tissues in healthy and inflammation models is not clear. As the clusters described in the first part of the paper (fig. 1b) are dominated by the tissues, a better experimental design could be used with corresponding tissues for healthy and inflammation models. Furthermore, the number of cells tested in the inflammation model was very low – is that the reason they could only find 39 genes induced upon inflammation? This seems a rather low estimate. Last, the inflammation analysis was lacking – could they find other neutrophil subsets or were these similar? again not exploiting the scRNA-seq data to its potential. In this form, a bulk measurement could be sufficient and even more informative for expression analysis.

3. Cluster 1, which is annotated as preNeu- this cluster is also composed of blood and spleen cells, which seems as a novel result. Are they proliferating (what is their G2M score)? Can the authors show them in separate than the BM cells in cluster 1? Can they be a source for more mature cells in the blood and the spleen? This result is an example for the power of single cell to discover rare population, but further experimental support is essential to indicate the significance of these findings. Maybe transplantation of sorted blood preNeu cells from mouse 45.1 into the blood of 45.2 mouse and follow up on these cells to see if they mature in the blood can answer this issue.

Other comments:

1. The scRNA-seq data is composed from 2 experiments that were joint together. However in fig. S1 k it seems that there are genes that deviate from the diagonal. Are they shared in all the tissues? Do they participate in a certain pathway? Are they removed from the analysis?

2. Figure 1C- the authors annotate the clusters in immature to mature scale; however it seems that only one gene is the reference to mature neutrophils (Ccl6). Are there other genes in this cluster support this annotation?

3. In p. 6:

'Cells from bone marrow resided at one end of this continuum, while neutrophils in blood and spleen clustered at the opposite end, suggesting again a spectrum based on maturation (Fig. 2b)' This cannot be understood from fig 2b. percentage of cells at each color rank or something similar should be added.

4. Figure 2d seems not in agreement with 2b, unless the authors will provide numbers/ percentages in 2b that will support the 2d.

5. Figure 7g- significant scores need to be added, and there is not visible difference and Cxcr2 expression in contrary to claim in the text.

General comments:

1. The figure legends need to be more self-explanatory and include more details (i.e number of mice in fig. 1a, the different panels in figure sup 4a etc.)

2. There are many figures referring which are not accurate. I.e- figures in the 'Neutrophils exhibiting a type I interferon signature' part (pg. 8), fig. 7c at the end of pg. 11 should be 7d etc.

3. There are genes mentioned in the paper in relation to a certain figure which do not appear in the figure itself, i.e.- Rpl32 (mentioned in pg. 4) in sup fig 2b, Cebpe (pg. 7) in fig 2F, Egr1 (pg. 12) in figure 8a

4. Ref 16,23 and 35 that appear in the discussion seem not relevant to the context they appear in (pg. 14)

5. The colors in fig 3a are not corresponding to the legend

6. Fig. 3b- are the cells ranges according to the neutrotime? Add scale to the X axis

7. The same for fig. 6a

8. Figure sup 1b- shown for the 23 clusters, while the rest of the analysis was done on 4 cluster model, this is confusing and damage the flow

REVIEWER COMMENTS

Reviewer #1 (neutrophils, cytokines, chemokines) (Remarks to the Author):

In this study, Grieshaber-Bouyer and colleagues applied single-cell RNA sequencing to murine neutrophils isolated from different biological tissues. The main result of the study is the identification of a neutrophil maturation trajectory, defined neutrotime. The study and all the bioinformatic analyses are accurate and well done. The major issues are related to the absence of comparative analysis in human neutrophils precursors and the lack of analysis of precursors, prior to PreNeu step, which would have allowed to validate the neutrotime even better.

RESPONSE: We thank the Reviewer for this encouraging assessment of our study design and analytical approach.

We agree that comparison with human neutrophils is important. Thus, we have obtained human scRNA-seq data from the Human Cell Atlas and analyzed cells from the neutrophil lineage. We were thereby able to reproduce the concept of neutrotime (presented in **new Fig. 5d-f**). These findings are discussed further in response to Reviewer 2. We consider the human/murine comparison a rich area for future investigation.

As described in Methods, we had intentionally restricted the analysis to cells at PreNeu and beyond using the ImmGen gene set data to exclude GMP. In response to the Reviewer's request, we returned to our primary data and were able to find 13 GMP cells that had been initially filtered out. Using RNA-Seq data from Kwok et al. *Immunity* 2020¹, we calculated a gene expression signature for proNeu1 and proNeu2 and applied it to the GMP cells. These findings, described below, support the Kwok results but do not otherwise impact the neutrotime thesis, which bears on the fate of neutrophils beginning at the preNeu stage.

Major points

1. On page 5, the authors point out that a broad correspondence between Monocle's pseudotime analysis and the clusters identified by UMAP is recognizable. This is true for P1 and probably for P4, but not for P2 and P3. In fact the P2 and P3 clusters identified by UMAP were composed mainly of BM cells while using Monocle P2 seems composed principally of spleen cells (extended Data Fig4a). The spleen is a hematopoietic organ in mice and one should expect the presence of a large number of neutrophil precursors, and therefore more in agreement with Monocle's analysis. Moreover, gating strategy to isolate neutrophils, as well as Ly6G and CD11b membrane expression from all three different sources (BM, blood and spleen), should be shown.

RESPONSE: We thank the Reviewer for this insightful comment. We have analyzed the clusters identified by UMAP (Seurat) and pseudotime results generated by Monocle again and find that P2 and P3 do in fact show a similar distribution in the Monocle analysis but this was not visible in the submitted figure due to overplotting. We have revised the figure with increased transparency and smaller points to show the distribution in UMAP split by source tissue. The fraction of cells by organ contributing to P1-P4 is independent of the low-dimensional representation of cells in UMAP generated by Seurat or Monocle, as the cluster identity of each cell was maintained for this analysis. The frequency of cells originating from each tissue is detailed in **Fig. 1e**.

As the Reviewer raised the interesting point about the frequency of precursors in the spleen, we quantitated preNeus across organs. We inspected the distribution of the preNeu score and chose 0.15 as cutoff to assign cells as preNeus (**new Reviewer Fig. 2**). Indeed, we find the majority of preNeus reside in bone marrow, accounting for 3.55% of sequenced neutrophils (**new Fig. 1i**). As anticipated by the Reviewer, preNeus were also found in spleen, accounting for ~ 1% of profiled neutrophils (**new Fig. 1i**). In contrast, only 5 cells assigned to the preNeu category by this cutoff were found in peripheral blood, representing 0.10% of profiled neutrophils. The identification of preNeus in the spleen in our scRNA-Seq dataset is in agreement with the flow cytometry analysis performed by Evrard and colleagues (*Immunity* 2018)³; however, those investigators found that preNeus "were absent from the blood (data not shown)". Our data suggest that preNeus may be found in peripheral blood in very small quantities, although their rarity limits options for detailed experimental study.

Continuing this analysis, we analyzed the G2M cell cycle score in preNeus across source tissues. We found that the G2M cell cycle score was significantly higher in preNeus compared to non-preNeus across all profiled tissues (**new Fig. 1j**), while no significant differences were found between preNeus across tissues, including peripheral blood (**new Fig. 1k**).

As requested by the Reviewer, we now present flow cytometry data and the gating strategy for all source tissues in **new Extended Data Fig. 1** and agree that this renders the findings more easily interpretable.

2 The indication that neutrophils exhibit an interferon signature is not evident from Figure 4. The authors should carefully revise the figure because there is no correspondence with the text. Only IFITM genes are differentially regulated along neutrotime and moreover not all in the same direction (Ifitm2 and Ifitm1 increased progressively with neutrotime while Ifitm6 was expressed preferentially early in neutrotime). All the other ISGs are not modulated therefore authors cannot refer to an interferon signature.

RESPONSE: We thank the Reviewer for this guidance. We have revised the text and changed the results heading to “**Expression of type I interferon-related transcripts along neutrotime**”. We agree that this phrasing is more precise. The revised text states:

“Neutrophils are exquisitely responsive to type I interferon, as reflected in the prominent transcriptomic impact of in vivo administration of this cytokine³⁵ (Fig. 4a). Neutrophils exhibiting transcriptional evidence of interferon response have been identified in models of cancer and inflammation, although not previously in healthy mice^{16,36}. Interestingly, neutrotime-defined maturation was prominently associated with a GSEA signature for type I interferon response (Fig. 3f). We thus examined expression of individual genes associated with type I interferon response along neutrotime (Fig. 4b). Ifitm2 and Ifitm1 increased progressively with neutrotime; Ifitm3 was expressed throughout neutrotime; and Ifitm6 was expressed preferentially early in neutrotime (Fig. 4c). By contrast, genes associated with type II interferon response were low in expression and exhibited no consistent pattern (Fig. 4d). An attempt to identify a discrete group of neutrophils by clustering on expression of type I response genes such as Ifit1, Ifit3, Isg15, and Ifitm3 failed to identify a subset of cells distinct from the neutrotime spectrum (Fig. 4e). Instead, genes associated with type I interferon response in neutrophils peaked in expression at different points along neutrotime (Fig. 4c and f), suggesting that the expression of interferon target genes in healthy neutrophils is a dynamic process that evolves with maturation. Whether this gene expression pattern arises through direct interferon exposure or by other means, and how interferon-related transcripts change neutrophil function, remains to be determined.”

3 In a recent study (PMID: 32579887) the group of Ng has identified two neutrophils precursors more immature than preNeu (proNeu1 and proNeu2). Authors should perform single-cell RNAseq also in this newly discovered cell types to fully validate their neutrotime model.

RESPONSE: We agree with the Reviewer that the analysis of neutrophil precursors upstream of preNeu is of potential interest. In our initial analysis step, cells were mapped to ImmGen reference populations and non-neutrophils (including GMP) were excluded. Accordingly, we re-analyzed our primary dataset for evidence of proNeu cells. Upon detailed inspection of the excluded precursors, we identified 13 granulocyte monocyte precursors (GMP), 4 multi lineage precursors (MLP) and 4 CD34+ hematopoietic stem cells in the dataset. We next obtained gene expression data corresponding to proNeu1 and proNeu2 from Kwok et al.¹ and calculated a gene expression signature corresponding to these cells using the same strategy as for preNeu marker genes in our original dataset. Heatmap representation of proNeu1 and proNeu2 marker genes in the identified progenitors revealed that proNeu1 and proNeu2 genes were more prominently expressed in GMPs compared to multipotent progenitors, supporting the proNeu paradigm proposed by Kwok and colleagues (**new Reviewer Fig. 1**). The limited sample size in our dataset did not allow us to assess definitively for different populations within GMPs. However, our study was not powered to address this point, which is not directly relevant for neutrotime, and so we reserve these data for Reviewers only.

4 Why Itf (lactoferrin), a specific granule protein, is included in the Combined ChEA3 and TF expression analysis of Figure 6e which should include only transcription factors. Authors should carefully check the list of transcription factors analyzed

RESPONSE: We appreciate the opportunity to clarify this result. *LTF/Ltf* is curated as a transcription factor in ChEA3, included in the co-expression libraries of ARCHS4 and GTEx. Lactoferrin has been shown to bind with high affinity to specific DNA motifs (GGCACTT(G/A)C, TAGA(A/G)GATCAAA, ACTACAGTCTACA)^{4,5}. The protein sequence of lactoferrin includes a nuclear localization signal suggested to be important in its transcriptional activity and highly conserved between mice and humans⁶. Thus, lactoferrin is a *bona fide* transcription factor as well as an anti-bactericidal secondary (specific) granule protein. Our analysis now adds to the understanding of lactoferrin by showing that its regulatory activity is associated with the transcriptome observed in early neutrotime. We have added these references and a paragraph related to this finding in the discussion. Further, we have reviewed the remaining list of transcription factors included to ensure that they are correct.

Minor points

1. HTOs were used also for datasets 1 and 2? Or different 10X runs were performed for blood, bone marrow and spleen. If this last is the case, how many cells were loaded per run?

RESPONSE: HTOs were used only in dataset 3 to tag the five separate experimental groups. Blood, bone marrow and spleen were always run in parallel, on the same 10X device, using one 10X lane for each tissue. In the first experiment, 6,000 cells were sorted and loaded per 10X lane (one separate 10X lane for each tissue), and recovery was between 20–26% per tissue. In the second experiment, 10,000 cells were loaded and recovery was between 38–47% per tissue. In the third 10X run, 12,000 cells total (hashed across five conditions as explained in the methods) were loaded, and recovery was 46%. We have added clarifying text to the methods.

2. Using bulk RNA-seq Evrad found that in the bone marrow are present mature neutrophils with similar characteristics to blood mature neutrophils. Instead, in the present study, using scRNA-seq authors find major differences (UMAP of figure 1d) and this should be at least discussed.

RESPONSE: We thank the Reviewer for highlighting this important point. We agree that the current interpretation of the Evrad *Immunity* 2018 data has been that it indicates the presence of a population of mature neutrophils in marrow that resemble those in blood. As the Reviewer notes, our data suggest that this interpretation requires a subtle revision. We observe that the two major increments in neutrotime are between preNeus and immature neutrophils, and between mature bone marrow neutrophils and mature neutrophils in blood (this finding is emphasized in **new Fig. 2g**). While this latter transcriptional shift by neutrotime-S is rather modest in degree (between 0.9450 and 0.9963; Fig. 5c) it is nevertheless quite clear, indicating that neutrophils in blood are meaningfully distinct from even the most mature subset in marrow.

To evaluate whether this conclusion is in fact in conflict with Evrad, we re-analyzed their bulk RNAseq data. As shown in **new Fig. 5d**, their dataset reveals substantial transcriptional differences between the most mature marrow neutrophils and blood (981 differentially expressed genes at log fold change ≥ 1 and adjusted $P < 0.05$), but the authors (correctly) observed that these shifts were much smaller than between GMPs and preNeus and between preNeus and mature neutrophils. The congruence between the Evrad data and our own is highlighted in **new Reviewer Fig. 3**.

We regard this observation as important to the field and have emphasized it within the text. We added new text to the Abstract, “The greatest increments in neutrotime were between preNeus and immature neutrophils and between mature marrow neutrophils and those in circulation.” In the Discussion, we now note “A sharp increase in neutrotime accompanies the transition from marrow to blood, suggesting that this transit event or other factors associated with circulation rapidly induce final neutrophil maturation.”

3 Which are the differentially expressed genes among neutrophils from blood or spleen? Even if minimal, is there an effect on gene expression of tissue environment?

RESPONSE: We appreciate this excellent suggestion. In response, we compared neutrophils at the same “stage” of neutrotime to examine tissue-specific signatures in blood, bone marrow, and spleen. To accomplish this, we binned together cells using a stepwise increase in neutrotime of 0.00001 until at least 5 cells per tissue and at least 30 cells total were represented in that bin. As the chosen interval was smaller than the smallest observable difference in neutrotime between two cells, this approach practically yielded a continuous collection of cells into a total of bins. We then calculated the average gene expression of neutrophils in each

respective bin, separately for each tissue. Visualization of the core neutrotime transcripts disclosed a remarkable similarity among expression profiles of key neutrotime genes, irrespective of tissue (**new Extended Data Fig. 8**). Differences were observed in only a few genes, such as *Ly6g* (almost no expression in peripheral blood). *Hbb-bs*, encoding Beta-globin, was detectable primarily in neutrophils from peripheral blood, potentially representing cell-free RNA from erythrocytes, as was recently described to be frequent in single cell experiments⁷.

Further, we modeled the expression of each gene as function of neutrotime and tissue using a linear model (neutrotime ~ tissue) followed by ANOVA. We found that for most genes, neutrotime explained the variance in expression better than did the source tissue. Exceptions included *Cd74*, which was detected primarily in spleen and simultaneously discovered by Hidalgo and colleagues in their recent multi-organ survey of neutrophils under homeostasis (Ballesteros et al. *Cell* 2020)⁸.

Together, these analyses suggest that neutrotime stage is responsible for the major fraction of the observed gene expression differences between neutrophils in addition to a small set of tissue specific programs.

4 In dot plots of fig1d, fig 4e (upper panel) and extended Data Fig4a (left panel) dots are too big and mask dots positioned below. For example, in extended Data Fig4a it seems that very few blood cells have been analyzed

RESPONSE: We thank the Reviewer for these comments. We tried a variety of visualization strategies to address this concern. We hope that Fig 1e and **new Fig. 2g** provides sufficient overview of the tissue contribution to different stages of neutrotime to eliminate the possibility of confusion.

5 Page 6: Extended Data Fig4d is mentioned in the text before Extended Data Fig4b

RESPONSE: We regret this error and have corrected it.

6 Fig. 3d-f: Authors state that: “Gene Set Enrichment Analysis (GSEA) similarly indicated enrichment of granule loading with early neutrotime and effector processes with late neutrotime (Fig. 3d-f)”. However, in the figure, the indication of the enrichment of granule loading should be better evidenced because absolutely not clear. Moreover, the most enriched gene sets are related to lymphocyte and not granule, this should be discussed

RESPONSE: We thanks the Reviewer for this suggestion. We have revised the figure to emphasize enrichment of granule loading in early neutrotime transcripts. We agree that GSEA can also yield unspecific terms, as indicated by the Reviewer. We further investigated the top hit by GSEA “HOFFMANN_LARGE_TO_SMALL_PRE_BII_LYMPHOCYTE_UP” to clarify this point. This gene set was first published in Hoffmann et al. with respect to B cells (Changes in gene expression profiles in developing B cells of murine bone marrow. *Genome Res.* 2002). However, many genes in this signature are not specific to lymphocytes but rather are common among actively proliferating cells. As preNeus contribute to the gene expression signature of early neutrotime, enrichment of this GSEA term is plausible. Additional clarification for the specificity of the enriched GSEA sets is provided by the GO term analysis presented in **Extended Data Fig. 5** which includes both generic metabolic terms (drug metabolic process, response to external stimulus, modification of morphology or physiology or other organism) as well as more specific terms associated with granule loading (defense response to bacterium, defense response to Gram-negative bacterium). We have added a clarification to the text that many genes in this signature are not specific to lymphocytes but rather common among actively proliferating cells and that neutrophil specific terms have been highlighted in the figure.

7 misspelling on page 13 last row: “are” instead of “ate”

RESPONSE: We regret this typographical error and have corrected it.

Reviewer #2 (granulocytes, neutrophils) (Remarks to the Author):

This paper identifies a chronologic developmental spectrum of neutrophil phenotypes, which the authors call neutrotime. Their data provide a strong argument for the notion that the various neutrophil subsets which can be identified actually reflect a single developmental continuum.

The study is well thought out, well executed, and very innovative. This being said, there remains a few discussion aspects that should be addressed (listed below), which would widen the scope of the findings.

RESPONSE: We thank the Reviewer for this kind assessment of the quality and significance of our study. We agree that the indicated discussion aspects further improve the scope and clarity of our findings.

1. The data focuses on gene expression patterns in various neutrophil subsets, but to which extent do these patterns relate to surface marker patterns observed in similar subsets, and in other well characterized neutrophil subsets (e.g. N1, N2), in vivo? There exist enough such data in the literature to start drawing parallels in the Discussion.

RESPONSE: We appreciate this suggestion. In the particular case of N1/N2 neutrophils (Fridlender et al. Cancer Cell. 2009) ⁹, transcriptomic data were not reported. However, a more recent paper (Zilionis et al. Cell 2019) ¹⁰ used single cell RNA-seq to investigate the myeloid compartment in human and murine samples with lung tumors. In their dataset, while they do partition neutrophils into a variable number of different clusters, these neutrophil states are densely interconnected as evident from their UMAP analysis. Upon detailed inspection, we do note several interesting parallels to our study. Importantly, several transcripts emerged from their analysis that were absent in ours, highlighting that while there is evidence that the neutrotime paradigm remains the underlying concept of heterogeneity in the tumor microenvironment, several environmental factors induce specific gene expression programs that are specific to the tumor environment.

We have considered how to address this question in our report. Unfortunately, in a paper that is already very long, we find that a detailed discussion comes at a substantial cost to our main focus, which is of a single continuum of neutrophils. Text that we drafted for the Discussion in response to the Reviewer's question, but ultimately decided NOT to include in the manuscript, is as follows:

“A large single cell survey of the myeloid cell compartment in lung tumors in both mice and humans [Zilionis] revised the previous N1/N2 classification using unbiased single-cell transcriptomics. In support of our neutrotime paradigm, the authors found a continuum of neutrophils both in humans and in mice which they then partitioned into 5 (human) and 6 (murine) neutrophil populations. As could be expected from changes to the tumor microenvironment, the transcriptome of these neutrophils did not simply mirror that found in our study across different organs; however, several similarities were observed. *Ngp* and *Chil3*, in our dataset expressed in very early neutrotime, were marked as “N₆” cells, suggesting that these are the least mature neutrophils in the dataset. *Chil3* is expressed in high abundance in preNeus, raising the possibility that preNeus also reside in the tumor microenvironment, and that the neutrotime paradigm persists in this environment. *Mmp8*, encoding a secondary granule protein, was associated with an intermediary stage of neutrotime and defined “N₁” cells. On the other hand, *Ccl3*, prominently expressed both in human as well as murine neutrophils termed “N₅”, was not found along unperturbed neutrotime. Instead, in our study *Ccl3* emerged as one of the transcripts which defined the conserved inflammatory response – a gene expression program shared in neutrophils across different inflamed scenarios. Similarly, *Cd14*, expressed as marker gene in “N₃” neutrophils, was only induced upon experimental inflammation. *Spp1* and *Id2*, two transcripts associated with “N₄” neutrophils, were again induced in experimental inflammation: *Spp1* was detected in great abundance in the inflamed joint and *Id2* in particular in IL-1 β induced peritonitis. *Ifit1*, marking the interferon responsive population in human and murine “N₂” neutrophils, was expressed at low abundance in our study, and clustering neutrophils based on interferon response genes did not give rise to a distinct population in our study. However, *Ifit1* was strongly induced in neutrophils harvested from the inflamed joint, again suggesting a marked effect of the inflamed environment on this population. We conclude that while a similar architecture to neutrotime can be observed in neutrophils from tumor microenvironments, we again find the neutrophil transcriptome greatly perturbed by environmental tissue, stimulus and time.”

With respect to surface markers, we conducted the experiment referenced in our comment to the Editors, wherein we compared expression of 8 relevant proteins by flow cytometry and scRNAseq (**new Extended Data Fig. 11**). These interesting new data highlight the well-recognized, but often underappreciated, differences between gene and protein expression. Identifying protein signatures of different neutrotime stages will require substantial work going forward.

2. Though the concept of neutrotime is likely to apply to human neutrophils as well, the authors should here again compare their data with patterns observed in corresponding or similar human neutrophil subsets. Because there exist so many striking differences between human and mouse neutrophils, where would one expect to see divergent patterns between the two species? This is obviously where this field of study is leading, so some discussion is warranted.

RESPONSE: We thank the Reviewer for this excellent suggestion. To provide additional parallels between the data presented in this manuscript and human data, we computed a list of transcripts that are associated with neutrotime that have one-to-one human orthologs with high confidence according to ENSEMBL version 100 (**new Supplementary Table 3**). Using these genes, we were able to generate a “humanized” signature of neutrotime. We then performed a detailed analysis of human neutrophil single cell data generated under the Human Cell Atlas. We included more than 7,000 bone marrow and cord blood cells from 8 human donors. In this analysis, we were able to find a strong gradient of the expression of early and late neutrotime genes and thus confirm a similar neutrotime signature (**new Fig. 5d-f**). Therefore, even though human and murine neutrophils have important differences, the core neutrotime paradigm appears to be conserved across species.

We had already noted in the manuscript that “later in neutrotime effector genes such as *Il1b* become predominant. These findings echo early studies of human neutrophils sorted at distinct developmental stages.” We have now noted the human correspondence in the Abstract as well as the Discussion. We believe that this additional validation in human data greatly widens the scope and significance of our findings.

Reviewer #3 (multiparameter flow, neutrophils) (Remarks to the Author):

In this paper, the Nigrovic group sets out to address an important and timely issue that is presently under intensive investigation in the “neutrophil community”. Neutrophil heterogeneity is receiving substantial attention and it remains to be fully established whether this heterogeneity stems from distinct ontogenetic developmental paths that are defined by distinct molecular programs or whether heterogeneity stems from plasticity of the respective neutrophil differentiation stages in response to environmental cues.

In this paper, the authors present a thorough single cell transcriptomic analysis of murine neutrophils under homeostatic conditions. They nicely outline the description of a differentiation program, that they decided to name “neutrotime”. In agreement with the authors, this reviewer agrees that the provided data sets and the data analysis represent a valuable resource for the research community. In the opinion of this reviewer, the connection and transition of homeostatic neutrotime to situations of inflammation, is less well-described in this otherwise excellent paper.

RESPONSE: We thank the Reviewer for this positive assessment and for the excellent suggestion for further improvement to this manuscript. Through additional analysis we now present our results of inflamed cells together with healthy cells in this substantially revised manuscript.

Conceptual critique:

Authors nicely outline in figure 1-6 the neutrotime transcriptional differentiation program in mice under non-inflamed homeostatic conditions. To this end, they investigate the neutrotime in blood, spleen and bone-marrow. In figure 7-8 the authors investigate neutrophils from blood, joint, lung and peritoneum of “arthritic” mice and mice exposed to Il-1b-driven inflammation.

Grieshaber-Bouyer et al. claim (in the abstract and other parts of the paper) that experimental inflammation translates into structured permutation of the neutrotime program. In the final part of the results section as well as in the final part of the discussion, the authors state and/or imply that during inflammation neutrophils follow the established neutrotime continuum. Variability under inflammatory conditions should then stem from the

varying extrinsic inflammatory factors that modulate mature neutrophils that reached this state through a common neutrotime program. With reference to the last sentence of the results section (page 12), a key question comes up: Do the authors show in this paper that the neutrotime continuum / programe is not perturbed and rather remains stable in the presence of inflammation, and in particular IL-1-driven inflammation?

When first reading the abstract, I expected to see an analysis of neutrotime, executed on bone-marrow, blood and spleen neutrophils under inflammatory conditions (in addition to naïve mice). Is neutrotime perturbed under inflammatory conditions? Why do the authors fail to show a direct comparison of neutrotime in naïve and inflamed mice.

Except for figure 7c, Fig 1-6 neutrotime/healthy is not well connected to Fig 7-8 inflammation.

RESPONSE: We appreciate this insight-provoking conceptual summary and address this issue in detail in response to critique 1 below.

Major critique:

1. Figure 2d shows that the vast majority of neutrophils in blood, BM and spleen belongs to two subpopulations that are located just before 0.25 and at the very end of the neutrotime scale. For the reader and people working in the field, it would be important to know, how this analyses would look like under conditions of inflammation.

RESPONSE: To address the Reviewer's comment, we performed a new analysis in which we synthesize data from non-inflamed and inflamed contexts.

We find that in a combined analysis, neutrophils harvested from inflamed sites (joint, peritoneum, lung) deviate markedly off the standard "arc" of neutrotime, whereas blood neutrophils from arthritic mice deviate only modestly (**new Fig. 7b**). However, essentially all neutrophils from inflamed tissues demonstrate an advanced neutrotime-S score (a simplified neutrophil index developed in the original manuscript that allows us to apply a numeric score based on expression of core neutrotime genes) comparable to blood, whereas almost none are early in neutrotime (**Fig. 7c**). This result is consistent with a model whereby these cells were mature blood neutrophils before they invaded inflamed tissues.

Next, we identified genes in neutrophils from inflamed sites that deviated from those in healthy tissues. These genes are shown in **Fig. 7d**. Many genes were found to be shared among all three inflamed sites, whereas others were shared only among the acute IL-1-driven models (lung, peritoneum), and still others appeared site-specific.

Visualization of such gene expression programs on the combined principal component analysis highlighted that early neutrotime genes remain downregulated in inflamed cells, while late neutrotime transcripts stay expressed throughout inflammation, suggesting again that the inflamed cells originate from blood cells advanced in neutrotime (**new Fig. 7e**).

Together, these studies show that the "core" neutrotime signature extends into the inflamed context, but that factors specific to the individual model (including site, stimulus, and time) drive the marked transcriptional deviation that distinguish these cells from resting blood neutrophils. Each neutrophil that migrates into tissue in response to an experimental stimulus begin at a discrete point within this continuum and then deviates transcriptionally as a function of environmental cues. We illustrate this principle with **new Fig. 9**.

We also expect that hematopoiesis will itself change under inflammatory conditions. For example, systemic IL-1 promotes myeloid skewing in the bone marrow (Pietras et al. *Nature Cell Biology*. 2016)¹¹. By contrast, in K/BxN serum transfer arthritis, changes in the peripheral blood are modest, as we see in the relatively subtle transcriptional differences between healthy and arthritic blood (**Fig. 7d**). Elucidation of the nature of these changes across a wide range of conditions will be an important future direction, but one distinct from the goal of this paper, which is to establish a single continuum of neutrophil development in the mouse and to provide evidence of a similar continuum in humans.

We have summarized this conclusion in the Discussion as follows:

“We thus propose a new working model (Fig. 9) to understand heterogeneity among neutrophils. Neutrophils represent a single lineage organized along one main maturational sequence, termed here neutrotime. As neutrophils progressing through this continuum encounter environmental cues, they deviate as a function of site, stimulus, and time, arriving at a phenotype that reflects both their starting position within neutrotime and the nature and sequence of signals encountered. This mechanism provides an opportunity to generate a highly diverse neutrophil repertoire without a requirement for committed developmental branches or cell subsets.”

2. Ext data figure 4c: It appears that neutrotime is not executed at “constant pace”. Rather two rapid maturation steps are included. Both are associated with the maturation of the two major subsets shown in figure 2d. This step-wise differentiation and maturation is not really emphasized in the text. Do I misinterpret the data or is this interpretation correct. If not correct, please explain in the rebuttal letter why not correct. If correct, shouldn't this be emphasized more in the manuscript?

RESPONSE: We agree with the Reviewer that there are two distinct increments in neutrotime: between preNeus and immature neutrophils, and between the most mature bone marrow neutrophils and those in blood. We have now added **new Fig. 2g** to illustrate these increments, and (as noted above in response to Reviewer 1, page 4) highlight the finding in both Abstract and Discussion. Whether this increment reflects “pace” per se (i.e. change over time) is more difficult to establish. Our RNA velocity data allow us to confirm that one cell stage comes before another; we are not aware of a way to determine the duration of each step.

3. Ext data figure 6: The transcriptional changes occurring in the 0.0 to 0.5 (first half) of neutrotime are nicely reflected in this figure. However, there is no cluster that is associated with the final steps of neutrophil maturation in the latest phase of neutrotime. Why is this so?

RESPONSE: This figure displays smoothed expression profiles of genes in the dataset where cells were ordered along neutrotime (from 0 to 1) and then randomly subsampled to 1000 cells for computation. Genes were clustered by k-means and centroids of each cluster were plotted in red. The scale on the x axis is the same as displayed in **new Fig. 2g**. Clusters 3 and 8 thus illustrate genes that increase in late neutrotime. Given the ordering of cells along neutrotime and the density of cells at the most mature pole, the graph gives the impression of few changes in the last half of the continuum. An analysis with cells binned together to detect tissue-specific differences (**new Extended Data Fig. 8**) also gives a better sense for the changes in the second half of neutrotime.

4. Ext data fig 6 – 8; and figure 2-4: Similar papers in the field combine RNASeq data and protein expression data. In this paper, we exclusively look at transcriptomic data. Is there a way that authors could translate some findings to the protein level? For example, transcriptomic data for chemokine receptors is important to show. However, surface expression of chemokine receptors is largely regulated by exposure to their ligands – including ligand-induced receptor downregulation. Second example; granule protein expression is only partly regulated on the transcriptomic level. Neutrophil activation and degranulation greatly influence the protein level of those components. In other words: Are we looking here at transcriptional profiles or are we also looking at biologically active protein profiles?

RESPONSE: The present manuscript seeks to use transcriptomes to define the ontological relationship among the diverse neutrophil phenotypes observed *in vivo*. For this purpose, the presence of surface or intracellular protein may even be misleading, since unlike mRNA (for which RNA velocity is available) protein has no standard signature indicating chronological order. As the Reviewer notes, the correlation between transcript abundance and protein abundance is limited, for reasons that include post-transcriptional regulation, protein stability, protein degradation, protein loss by secretion or cleavage from the cell surface, and others. Correspondingly, the correlation between transcript and protein is at best no greater than than 0.43¹². An excellent example from our analysis is the canonical murine neutrophil surface marker Ly6G, which at the protein level is present at highest levels in mature neutrophils and which we use here for cell isolation, yet for which mRNA is highest in early neutrotime and essentially absent in the most mature cells. For this reason, we make no claim here to biologically active protein profiles.

However, we agree that the Reviewer's point is important and will be of interest to readers. Therefore, we repeated each experimental condition to create flow cytometry data corresponding to proteins encoded by 8

genes of substantial variability and biological interest, including chemokine receptors as noted by the Reviewer (*Cd9, Cd14, Cd53, Cd63, Itgam, Cxcr2, Cxcr4, Ly6g*) (**Flow cytometry panels: Supplementary Table 2**). We find that the correlation between gene and protein is greatly dependent on the context, highlighting the important point that protein level cannot be safely extrapolated from gene expression (**Extended Data Fig. 11**). We have amended the manuscript to highlight these data, in particular with respect to Ly6G and the chemokine receptors. In future studies, we look forward to using CITE-seq to identify protein signatures for each stage of neutrotime and thereby to further explore the functional implications of this maturational continuum.

Minor issues:

1. Ext data Fig 1: what are the Ly6G negative cells in panel B? According to panel A, only Ly6G positive cells are within the gate?

RESPONSE: Extended Data Fig. 1b shows neutrophils stained with either anti-Ly6G or isotype control and either anti-CD11b or isotype control. The blue histograms show expression in the antibody-stained group, while the gray histograms show the corresponding isotype controls. In this case, expression for all cells in the FSC-SSC gate were shown. Thus, we believe that monocytes may have constituted the peak of Ly6G-negative cells. Our **Extended Data Fig. 1** has been substantially revised and now includes gating strategies for all tissues as described in response to the next comment.

2. Expression of Ly-6G and CD11b differs between blood, spleen and BM neutrophils (and lung/peritoneum/joint neutrophils) and between naïve and inflamed mice. As these gates and FACS are the starting point for all analyses, it would be nice to extend the “ext data fig 1a” accordingly and show all starting populations.

RESPONSE: Our **revised Extended Data Fig. 1** now shows the gating strategy for all different tissues.

3. Color code of P1-4 does not seem to match the populations shown in UMAP plots.

RESPONSE: We thank the Reviewer for highlighting this point. We believe that the hue shift may have arisen from different color profiles in vectorized or rasterized graphics in Adobe Illustrator. We have revised the Figure so that the color code matches the populations in the UMAP plots and will supply vector images to the production team to optimize color reproduction for publication.

Reviewer #4 (single cell analysis, RNAseq) (Remarks to the Author):

The manuscript by Grieshaber-Bouyer et al aims to study neutrophil subsets in health and disease. Understanding Neutrophil heterogeneity and how different neutrophil subsets are functionally distinct is a timely and important question. To achieve this, the authors use scRNA-seq to decipher cell subsets based on their transcriptional profiles. Importantly, this method requires elaborative analysis to achieve its potential, as well experimental follow up to validate the reliability of the findings. Unfortunately, the data presented in this paper, of 4 main subsets, fails to expand our knowledge as these exact findings were already reported (DOI:<https://doi.org/10.1016/j.immuni.2018.02.002>). Furthermore, the data fails also to discriminate between neutrophil subsets in different tissues in health and disease (e.g., bone marrow blood and so on). Last, the authors do not provide functional experiments to support their scRNA-seq analysis.

RESPONSE: We thank the Reviewer for these comments. We highlight however that our data *reject* the hypothesis of neutrophil subsets, showing instead that murine neutrophils form a single continuum. This continuum may indeed be split into 4 subsets, or 9 subsets, or 23 subsets, depending on modeling assumptions (**Extended Data Fig. 3**) – however, as we show, these clusters represent “time slices” of a single kind of neutrophil. This conclusion is arrived at through transcriptomic analysis because of the distinct advantages of this approach, in particular the time vector afforded by RNA velocity. By contrast, as highlighted in the response to Reviewer 3, protein signatures (and by extension also functional studies) remain fundamentally ambiguous in terms of ontologic relationships among cells. While we expect that cells at different stages of the neutrotime continuum will have different functions, we make no specific claims to that effect here that might benefit from the functional experiments described. In our revised manuscript, we provide a detailed analysis to address the question how much of the identity of a neutrophil is described as

a function of neutrotime stage versus source tissue. In **new Extended Data Fig. 8** we show that the vast majority of variance arises from neutrotime stage and not source tissue.

Major comments:

1. Using scRNA-seq, the authors aim to understand neutrophil subsets within different organs. The authors chose the 4-cluster model (fig. 1b) as their main classifier of neutrophil subsets in the paper. However, similar subsets that define developmental trajectory and functional properties of neutrophils in bone marrow, blood and tissues was already studied (see the DOI above). Thus, the finding of 4 clusters is not novel. It is surprising that the authors did not look for possible differences in models with higher cluster number (sup figure 2a) or further apply knn on each cluster to look for delicate changes between subpopulations in each cluster. The scRNA-seq analysis, which is the focus of the paper, is not comprehensive.

RESPONSE: We regret any confusion on this point. We employed 4 clusters explicitly as a tool to exhibit the developmental orientation of the genes that vary sequentially across neutrotime. As we note explicitly in the text, the choice to use this model was arbitrary and selected for convenience. We do not attempt to reify these as subtypes, and indeed we show that the numbers of clusters is unstable across a range of reasonable modeling assumptions, including the number of neighbors considered during clustering, leading us to propose neutrotime as an alternative. **It is this rejection of the subset model that represents the most novel and important conclusion of our study.** We are uncertain as to the additional scRNA-seq analyses that the Reviewer would have considered necessary for comprehensiveness; we note the Reviewers 1, 2 and 3 had not shared this concern.

2. Even more confusing is the scRNA-seq analysis of the inflammation models. First, the choice of different tissues in healthy and inflammation models is not clear. As the clusters described in the first part of the paper (fig. 1b) are dominated by the tissues, a better experimental design could be used with corresponding tissues for healthy and inflammation models. Furthermore, the number of cells tested in the inflammation model was very low – is that the reason they could only find 39 genes induced upon inflammation? This seems a rather low estimate. Last, the inflammation analysis was lacking – could they find other neutrophil subsets or were these similar? again not exploiting the scRNA-seq data to its potential. In this form, a bulk measurement could be sufficient and even more informative for expression analysis.

RESPONSE: We thank the Reviewer for these comments. In response, as outlined in response to Reviewer 2, we introduce new **Figure 7** that illustrates both the neutrotime continuity and the model-specific transcriptional changes that occur at the single-cell level. As the summary plot demonstrates, each inflammatory model is represented by hundreds of individual cells, providing ample power to identify significant differences between models and between sites, even with the same pro-inflammatory trigger (IL-1 β peritonitis vs. IL-1 β pneumonitis). We suspect that the Reviewer's concern about our power was focused on the K/BxN blood vs. healthy blood comparison, wherein there were indeed only 39 genes that cross the significance threshold of FDR 0.01 and a log fold change of at least 0.25. However, we emphasize that this model is one of joint inflammation, with relatively minimal systemic inflammation. The number of analyzed cells was more than sufficient to detect much larger gene expression changes in other comparisons. Using the same threshold as above, compared to healthy blood, we found 675 differentially expressed genes for IL-1 pneumonitis, 657 genes for IL-1 peritonitis, and 451 genes in K/BxN arthritis joint tissue. The fact that only 39 genes differentiate K/BxN blood neutrophils from those in healthy blood serves to underscore the fact that inflammation in this model is highly focused on the joint.

3. Cluster 1, which is annotated as preNeu- this cluster is also composed of blood and spleen cells, which seems as a novel result. Are they proliferating (what is their G2M score)? Can the authors show them in separate than the BM cells in cluster 1? Can they be a source for more mature cells in the blood and the spleen? This result is an example for the power of single cell to discover rare population, but further experimental support is essential to indicate the significance of these findings. Maybe transplantation of sorted blood preNeu cells from mouse 45.1 into the blood of 45.2 mouse and follow up on these cells to see if they mature in the blood can answer this issue.

RESPONSE: We agree that the discovery of preNeus in the circulation is an interesting finding. However, the paucity of these cells in the blood (5 cells in our sample; **new Fig. 1i**) render adoptive transfer of blood cells impractical, whereas transfer of these cells from spleen (where they represent 1% of neutrophils) would

be uninformative since the spleen is well recognized to be a hematopoietic organ, such that transfer would simply replicate the work of Evrard et al. As suggested, we determined the G2M scores of preNeu in marrow, spleen and blood and found that preNeus from all organs were actively cycling, indicated by a high G2M score (**new Fig. 1j**). We did not find statistically significant differences in the G2M cell cycle score across tissues (**new Fig. 1k**). This result is detailed in response to Reviewer 1 and is included in the paper with new text in Results and Discussion.

Other comments:

1. The scRNA-seq data is composed from 2 experiments that were joint together. However in fig. S1 k it seems that there are genes that deviate from the diagonal. Are they shared in all the tissues? Do they participate in a certain pathway? Are they removed from the analysis?

RESPONSE: ScRNA-Seq studies performed on different days and with different 10X chemistries inevitably show variance in gene expression. We found that the correlation between experimental repeats was very high (see Extended Data Fig. 1 of the original submission) and therefore could analyze cells together. To address the Reviewer's concern, we performed differential expression analysis between the experiments, separately for each tissue. Using the same thresholds as throughout the manuscript (FDR 0.01, log fold change 0.25), we found 14 differentially expressed genes in peripheral blood, 60 in bone marrow and only 1 in spleen between the two experiments, highlighting that gene expression was very similar in both datasets. Interestingly, some of the differentially expressed genes overlapped with the core neutrotime signature. To assess whether these genes may have impacted downstream analysis, we split our dataset by tissue and experiment, ordered cells along neutrotime and analyzed expression of the differentially expressed genes. We found remarkable similarity in the expression dynamics of the differentially expressed genes between both datasets (**new Reviewer Fig. 4**) and conclude that the observed small dataset-specific differences did not affect global data architecture. We believe that this figure does not add to the understanding of the manuscript and thus reserve it as Reviewer only.

2. Figure 1C- the authors annotate the clusters in immature to mature scale; however, it seems that only one gene is the reference to mature neutrophils (*Ccl6*). Are there other genes in this cluster support this annotation?

RESPONSE: We thank the Reviewer for this opportunity to clarify the text. In addition to *Ccl6*, *Csf3r*, encoding the G-CSF receptor, is also associated with mature neutrophils. Furthermore, *Il1b* (*IL1B*) was found in humans in the most mature bone marrow neutrophils (Theilgaard-Monch et al. Blood 2005)¹³ and in the present study we found high *Il1b* expression in peripheral blood neutrophils. Note that prior single-cell analyses of neutrophils had not noted the striking elevation of these transcripts in blood neutrophils through their focus on bone marrow progenitors and immature neutrophils (e.g. Giladi et al. 2018)¹⁴, which according to our neutrotime model do not yet express *Il1b*. We have added these transcripts to the text as follows: and progressing in Cluster 4 to genes associated with mature neutrophils, such as *Csf3r*, encoding the G-CSF receptor, *Il1b*, encoding interleukin-1 β and the chemokine-encoding gene *Ccl6* (Fig. 1c).

3. In p. 6:

'Cells from bone marrow resided at one end of this continuum, while neutrophils in blood and spleen clustered at the opposite end, suggesting again a spectrum based on maturation (Fig. 2b)'

This cannot be understood from fig 2b. percentage of cells at each color rank or something similar should be added.

RESPONSE: We agree with the Reviewer than quantitation is important here. In **new Figure 2g** we show a detailed analysis of the frequency of each tissue contributing to different points along neutrotime and highlight striking increments of neutrotime at two points (preNeu to immature neutrophil and bone marrow to blood transition). We believe that this scale helps interpret the findings.

4. Figure 2d seems not in agreement with 2b, unless the authors will provide numbers/ percentages in 2b that will support the 2d.

RESPONSE: We agree with the Reviewer and have therefore added **new Fig. 2g** which shows the density of cells along neutrotime and along the maturation score for each tissue as well as quantiles of neutrotime and cell percentages.

5. Figure 7g- significant scores need to be added, and there is not visible difference and *Cxcr2* expression in contrary to claim in the text.

RESPONSE: We thank the Reviewer for this suggestion and have revised the figure to now show an alternative visualization of the *Cxcr2* and *Cxcr4* expression in each condition. We have also added the median expression (diamond) and the percentage of cells with non-zero expression (gray bars) and added statistics. We believe that this clarification highlights that the expression differences in *Cxcr2* and *Cxcr4* are in line with the description in the text.

General comments:

1. The figure legends need to be more self-explanatory and include more details (i.e number of mice in fig. 1a, the different panels in figure sup 4a etc.)

RESPONSE: We thank the Reviewer for this comment and have revised the figure legends to include more details such as the number of mice, number of experiments and detailed description of each panel.

2. There are many figures referring which are not accurate. I.e- figures in the 'Neutrophils exhibiting a type I interferon signature' part (pg. 8), fig. 7c at the end of pg. 11 should be 7d etc.

RESPONSE: We regret this mistake and have carefully revised the manuscript and the figures to align.

3. There are genes mentioned in the paper in relation to a certain figure which do not appear in the figure itself, i.e.- *Rpl32* (mentioned in pg. 4) in sup fig 2b, *Cebpe* (pg. 7) in fig 2F, *Egr1* (pg. 12) in figure 8a

RESPONSE: We thank the Reviewer for drawing our attention to these important details. We found that in fact *Cebpe* appears in Fig. 3b and the validation in 5a but not in 2F – likely arising from different analysis and manuscript versions with different thresholds for the gene lists. Th same goes for *Egr1*, which was thus removed from the text. *Rpl32* as also mentioned by the reviewer appears in **Extended Data Fig. 2b (now 3b)**. We have carefully revised text and figures so that they align in the final version.

4. Ref 16,23 and 35 that appear in the discussion seem not relevant to the context they appear in (pg. 14)

RESPONSE: We have carefully revisited the Discussion and checked the context in which they appear. In the example highlighted by the Reviewer, we had stated that "Prior single-cell transcriptional studies of neutrophils have focused either on early development or within the context of tumor or inflammation" and cited work by Zilionis et al. (tumor microenvironment) and Radermecker et al. (allergic airway inflammation). We agree that the reference to the study performed by Evrard et al. (bulk RNA-seq) is not relevant in this context and have removed it. The other references appear to match the context well in which they are presented.

5. The colors in fig 3a are not corresponding to the legend

RESPONSE: We regret this error and have changed the color of the legend representing non correlated transcripts to match the data points.

6. Fig. 3b- are the cells ranges according to the neutrotime? Add scale to the X axis

7. The same for fig. 6a

RESPONSE to 6 and 7: Cells are indeed ordered by neutrotime in this figure. We have added a scale for the X axis in new Figure 2g as suggested, which applies to other figures in which cells are ordered along neutrotime.

8. Figure sup 1b- shown for the 23 clusters, while the rest of the analysis was done on 4 cluster model, this is confusing and damage the flow

RESPONSE: Gene expression on a per-cluster basis is shown for the 4-population model in **Fig. 1c**. The goal of the supplemental figure was to show the limitations of the clustering approach, additionally presenting gene expression for the clustering with the highest number of clusters. For this reason, we introduce neutrotime – a continuous model – abandoning clusters entirely beginning in Figure 2.

Again, we thank the Reviewers and the Editors for their detailed attention to our manuscript, which has allowed us to improve the work. We look forward to your consideration of the revised work.

Sincerely yours,

Peter A. Nigrovic, MD for the authors

References

1. Kwok, I. *et al.* Combinatorial Single-Cell Analyses of Granulocyte-Monocyte Progenitor Heterogeneity Reveals an Early Uni-potent Neutrophil Progenitor. *Immunity* (2020). doi:10.1016/j.immuni.2020.06.005
2. Muench, D. E. *et al.* Mouse models of neutropenia reveal progenitor-stage-specific defects. *Nature* 1–29 (2020). doi:10.1038/s41586-020-2227-7
3. Evrard, M. *et al.* Developmental Analysis of Bone Marrow Neutrophils Reveals Populations Specialized in Expansion, Trafficking, and Effector Functions. *Immunity* **48**, 364–379.e8 (2018).
4. Mariller, C. *et al.* Human delta-lactoferrin is a transcription factor that enhances Skp1 (S-phase kinase-associated protein) gene expression. *FEBS J.* **274**, 2038–2053 (2007).
5. He, J. & Furmanski, P. Sequence specificity and transcriptional activation in the binding of lactoferrin to DNA. *Nature* **373**, 721–724 (1995).
6. Mariller, C. *et al.* Delta-lactoferrin, an intracellular lactoferrin isoform that acts as a transcription factor. *Biochem Cell Biol* **90**, 307–319 (2012).
7. Young, M. D. & Behjati, S. SoupX removes ambient RNA contamination from droplet based single-cell RNA sequencing data. *bioRxiv* 303727 (2020).
8. Ballesteros, I. *et al.* Co-option of Neutrophil Fates by Tissue Environments. *Cell* (2020). doi:10.1016/j.cell.2020.10.003
9. Fridlender, Z. G. *et al.* Polarization of tumor-associated neutrophil phenotype by TGF-beta: "N1" versus 'N2' TAN. *Cancer Cell* **16**, 183–194 (2009).
10. Zilionis, R. *et al.* Single-Cell Transcriptomics of Human and Mouse Lung Cancers Reveals Conserved Myeloid Populations across Individuals and Species. *Immunity* **50**, 1317–1334.e10 (2019).
11. Pietras, E. M. *et al.* Chronic interleukin-1 exposure drives haematopoietic stem cells towards precocious myeloid differentiation at the expense of self-renewal. *Nat Cell Biol* **18**, 607–618 (2016).
12. Eastman, G., Smircich, P. & Sotelo-Silveira, J. R. Following Ribosome Footprints to Understand Translation at a Genome Wide Level. *Comput Struct Biotechnol J* **16**, 167–176 (2018).
13. Theilgaard-Mönch, K. *et al.* The transcriptional program of terminal granulocytic differentiation. *Blood* **105**, 1785–1796 (2005).
14. Giladi, A. *et al.* Single-cell characterization of haematopoietic progenitors and their trajectories in homeostasis and perturbed haematopoiesis. *Nat Cell Biol* **20**, 836–846 (2018).

Reviewer Figure 1 | Gene expression signatures of proNeu 1 and proNeu 2 in precursors in the dataset.

A module score was calculated for a proNeu 1 and proNeu 2 gene expression signature for precursors which had initially been filtered out of the dataset.

Reviewer Figure 2 | Distribution of preNeu score across cells.

The distribution of the module score for the preNeu gene set and the chosen cutoff to assign cells the preNeu category is shown.

Reprinted from Immunity, Volume 48, Issue 2,
Evrard, Maximilien et al, 364 - 379.e8,
Copyright 2018, with permission from Elsevier

Evrard et al. 2018

Grieshaber Bouyer et al. 2020

Reviewer Figure 3 | Comparison of the proximity of populations in Evrard et al. 2018 and neutrotime (this paper).

Left panel: original principal component analysis by Evrard et al.

Right panel: Mapping of the Evrard et al. bulk RNA-Seq data on Neutrotime-S

Reviewer Figure 4 | Heatmaps showing expression of differentially expressed genes between datasets 1 and 2.

N number for each tissue and heatmap indicates the number of differentially expressed genes in the respective compartment. Cells were sorted along neutrotime for visualization.

REVIEWER COMMENTS

Reviewer #1 (Remarks to the Author):

The authors have fully addressed my concerns. Very good job

Reviewer #2 (Remarks to the Author):

I have read with great interest the revised manuscript by Grieshaber-Bouyer et al.

I am satisfied to report that the authors duly addressed the points that I had raised. I therefore recommend that this excellent paper be accepted for publication.

Reviewer #3 (Remarks to the Author):

The authors have substantially improved the manuscript. They have also added a schematic figure 9 that intends to illustrate the main conclusion and argument that we should derive from this paper.

I have the following remaining critiques that are required in order to improve the general understandability of the data also to the non-expert (this journal has a broad readership).

Major Critique 1:

The authors have clearly improved the incorporation of "inflamed" neutrophils into their neutrotime paradigm. They have also included a more direct comparison of neutrophils from "inflamed" mice with neutrophils from healthy mice. While these new data are included in the results section, the implications of these findings are not yet sufficiently mentioned in the discussion. To clarify the "healthy vs inflamed comparison" to the reader the discussion should also be refined.

The point that neutrophils in inflamed tissues largely derive from mature cells and that only very few (if any) immature cells or progenitor or Pre-Neu cells enter the site of inflammation needs to be stated more clearly in the Ms.

I suggest to include at least one aspect that is clearly included in the rebuttal letter, but not yet in the discussion:

However, essentially all neutrophils from inflamed tissues demonstrate an advanced neutrotime-S score (a simplified neutrophil index developed in the original manuscript that allows us to apply a numeric score based on expression of core neutrotime genes) comparable to blood, whereas almost none are early in neutrotime (Fig. 7c). This result is consistent with a model whereby these cells were mature blood neutrophils before they invaded inflamed tissues.

early neutrotime genes remain downregulated in inflamed cells, while late neutrotime transcripts stay expressed throughout inflammation, suggesting again that the inflamed cells originate from blood cells advanced in neutrotime

Major critique 2:

abstract: I am afraid that I substantially disagree with this sentence of the abstract:

"Neutrophils migrating into inflamed lung, peritoneum and joint exhibited de novo transcriptional activity representing structured permutations of neutrotime."

Firstly, it does not state the key findings of the experimental data. According to the data it seems that only neutrophils at the very advanced stages, with a high neutrotime-S-score, are found in tissues.

Secondly, I am afraid that the term "structured permutation" is misleading here.

In mathematics, according to the original definition of the term, a permutation is an arrangement of its members into a sequence of LINEAR order.

In this example the neutrophil subsets: P1-P2-P3-P4

For each set of n distinct members, the number of permutations is n! (for four populations this would result in $4 \times 3 \times 2 \times 1 = 24$ possible permutations)

In these permutations, the members can alternatively appear in different sequences or orderings. For example, P1-P3-P4-P2

However, in this paper, inflammation does NOT change the linear order of the P1-P4 neutrotime elements but rather results in a DEVIATION to the side as nicely illustrated in figure 9.

The phrase used in the rebuttal letter more accurately reflects these findings:

Together, these studies show that the "core" neutrotime signature extends into the inflamed context, but that factors specific to the individual model (including site, stimulus, and time) drive the marked transcriptional deviation that distinguish these cells from resting blood neutrophils. Each neutrophil that migrates into tissue in response to an experimental stimulus begin at a discrete point within this continuum and then deviates transcriptionally as a function of environmental cues.

As a final note: In the paper, the authors primarily show the deviation of mature neutrophils with "high" neutrotime score.

Minor Critique 1:

Figure 9: please modify the dashed lines to improve visibility. Did not see it when I first looked at the figure.

Minor critique 2:

Figure 9: this figure is a nice and valuable addition to illustrate the core principle of the paper. However, according to the presented data, the deviation from the "core" neutrotime programme as a consequence of inflammation does mainly occur at the mature end of the neutrotime scheme (high neutrotime-S score). This should be better illustrated in the figure. As it stands now, it seems that deviations into the direction of inflammatory tissue neutrophils equally occur at all stages of neutrotime. As far as I understood the paper, this is not the case and not supported by the data.

With regards

Sven Brandau

Reviewer #4 (Remarks to the Author):

In their revised version, the authors added new analysis (i.e. human neutrophils analysis, mRNA-protein comparison) and clarified the overall message of the paper. However, my previous criticism remains unanswered. I'm very concerned about the novelty of the manuscript and the support for the main claim: 'It is this rejection of the subset model that represents the most novel and important conclusion of our study'. I am afraid that the authors do not provide ample evidence to support their claim. Moreover, the authors claim in the letter to the reviewers that 'While we expect that cells at different stages of the neutrotime continuum will have different functions, we make no specific claims to that effect here that might benefit from the functional experiments described'. It is my opinion that the lack of any functional experiments to support their claim leaves this study very descriptive and do not add useful information to the field.

A non-comprehensive list of comments is below.

1. The fact that the authors apply Monocle and diffusion maps algorithms to their data, that display a continuum is not surprising. This is the heart of these algorithms. Many other datasets can be depicted similarly, but the real challenge is to show that indeed this data representation can be validated. Otherwise, strong claims made by the authors remain ambiguous and, unfortunately, not concise.
2. Furthermore, in light of recent publications, especially- <https://doi.org/10.1038/s41590-020-0736-z> (cited in the manuscript), rejection of the subset model should be better discussed. In the mentioned paper, the exact tissues were studied in mice, revealing a trajectory of subtypes of neutrophils. Their model was then validated in the context of infection and transcription factors, further questioning the novelty of the current manuscript.
3. The authors present an alternative model here, the continues model for neutrophils. However, the results themselves seems not to support this model:
 - a. The claim for 'instability of the clusters' cannot be deduced from the results in supp fig. 2a. Obviously, changes of the parameters will affect the number of clusters. However, P1-3 seems very distinct.
 - b. It will be hard to convince the TF data in Figure 6 follows a continuum rather than subsets.

Same claim to the IFN-I data. If the authors want to claim continuum, they need to provide evidence.

c. The validation of the neutrotime model (fig. 5) does not validate a continues pattern. First, the correlation between neutrotime and neutrotime-s genes seems problematic. Despite the high Spearman correlation coefficient, there are ranges that are dramatically modified. For example, the whole range of 0.25-0.75 in neutrotime score will get a very small score range in neutrotime-s of ± 0.75 . Furthermore, the bulk studies do support different expression of the neutrotime-s genes, however the validation should be to continues expression, not to the expression of the selected genes.

d. In figure 2G, the authors define a bone marrow to blood transition: "Blood neutrophils in healthy mice overwhelmingly occupy the most mature end of the spectrum. A sharp increase in neutrotime accompanies the transition from marrow to blood, suggesting that this transit event or other factors associated with circulation rapidly induce final neutrophil maturation." What is the support for this with a continuum as opposed to clustered data?

e. In their infection model, the authors use the neurotime genes. Where is the advantage of a continuum analysis here? Genes from P1-P4 seem to have been just as relevant.

Reviewer #1:

The authors have fully addressed my concerns. Very good job

RESPONSE: We thank the Reviewer for this positive assessment of the revised work.

Reviewer #2:

I have read with great interest the revised manuscript by Grieshaber-Bouyer et al.

I am satisfied to report that the authors duly addressed the points that I had raised. I therefore recommend that this excellent paper be accepted for publication.

RESPONSE: We thank the Reviewer for this positive assessment of the revised work.

Reviewer #3:

The authors have substantially improved the manuscript. They have also added a schematic figure 9 that intends to illustrate the main conclusion and argument that we should derive from this paper.

I have the following remaining critiques that are required in order to improve the general understandability of the data also to the non-expert (this journal has a broad readership).

Major Critique 1:

The authors have clearly improved the incorporation of “inflamed” neutrophils into their neutrotime paradigm. They have also included a more direct comparison of neutrophils from “inflamed” mice with neutrophils from healthy mice. While these new data are included in the results section, the implications of these findings are not yet sufficiently mentioned in the discussion. To clarify the “healthy vs inflamed comparison” to the reader the discussion should also be refined.

The point that neutrophils in inflamed tissues largely derive from mature cells and that only very few (if any) immature cells or progenitor or Pre-Neu cells enter the site of inflammation needs to be stated more clearly in the Ms.

I suggest to include at least one aspect that is clearly included in the rebuttal letter, but not yet in the discussion:

However, essentially all neutrophils from inflamed tissues demonstrate an advanced neutrotime-S score (a simplified neutrophil index developed in the original manuscript that allows us to apply a numeric score based on expression of core neutrotime genes) comparable to blood, whereas almost none are early in neutrotime (Fig. 7c). This result is consistent with a model whereby these cells were mature blood neutrophils before they invaded inflamed tissues.

early neutrotime genes remain downregulated in inflamed cells, while late neutrotime transcripts stay expressed throughout inflammation, suggesting again that the inflamed cells originate from blood cells advanced in neutrotime

RESPONSE: We appreciate Dr. Brandau’s suggestion to enhance the clarity of the work. We have revised the Discussion to better incorporate the data from inflamed animals. We note in particular that “Importantly, neutrophils recruited to sites of inflammation were generally far advanced along the neutrotime spectrum, suggesting that these cells were mature blood neutrophils before tissue entry.”

Major critique 2:

abstract: I am afraid that I substantially disagree with this sentence of the abstract:

“Neutrophils migrating into inflamed lung, peritoneum and joint exhibited de novo transcriptional activity representing structured permutations of neutrotime.”

Firstly, it does not state the key findings of the experimental data. According to the data it seems that only neutrophils at the very advanced stages, with a high neutrotime-S-score, are found in tissues.

Secondly, I am afraid that the term “structured permutation” is misleading here.

In mathematics, according to the original definition of the term, a permutation is an arrangement of its members into a sequence of LINEAR order.

In this example the neutrophil subsets: P1-P2-P3-P4

For each set of n distinct members, the number of permutations is n! (for four populations this would result in $4 \times 3 \times 2 \times 1 = 24$ possible permutations)

In these permutations, the members can alternatively appear in different sequences or orderings.

For example, P1-P3-P4-P2

However, in this paper, inflammation does NOT change the linear order of the P1-P4 neutrotime elements but rather results in a DEVIATION to the side as nicely illustrated in figure 9.

The phrase used in the rebuttal letter more accurately reflects these findings:

Together, these studies show that the “core” neutrotime signature extends into the inflamed context, but that factors specific to the individual model (including site, stimulus, and time) drive the marked transcriptional deviation that distinguish these cells from resting blood neutrophils. Each neutrophil that migrates into tissue in response to an experimental stimulus begin at a discrete point within this continuum and then deviates transcriptionally as a function of environmental cues.

As a final note: In the paper, the authors primarily show the deviation of mature neutrophils with “high” neutrotime score.

RESPONSE: We have revised further to address these comments.

First, although the tissue neutrophils do all appear to stem from the mature end of neutrotime, the relative immaturity of blood neutrophils in inflammatory contexts suggests the possibility that neutrophils might develop distinctive phenotypes upon encountering environmental stimuli at any stage of neutrotime. This possibility is reflected in Figure 9. We have clarified this point further in the text as follows: “For neutrophils recruited to inflamed sites in the models studied here, these cells primary emerge from the mature end of the spectrum, but the relative immaturity of circulating cells observed during inflamed states suggests that similar phenotypic divergence could occur at any stage of neutrotime.” We added to the legend of Figure 9, “Experimental inflammation was found to recruit neutrophils near the mature pole of neutrotime; however, deviation from points earlier in the spectrum is also likely, reflected in arrows all along the neutrotime continuum.”

Second, we appreciate the suggestion with respect to “permutation”. In general usage this term does not imply linearity or rearrangement of parts; for example, “technology available in various permutations” is a phrase cited by Merriam-Webster to exemplify correct use, but would fail Dr. Brandau’s mathematics-derived standard. Nevertheless, we agree that a more descriptive phrase would be better and have adopted the following per the Reviewer’s suggestion: “Neutrophils migrating into inflamed lung, peritoneum and joint maintained core mature neutrotime genes but exhibited substantial de novo transcriptional activity depending on site and stimulus.”

Minor Critique 1:

Figure 9: please modify the dashed lines to improve visibility. Did not see it when I first looked at the figure.

RESPONSE: Both lines have been intensified and the dashed lines darkened and colored for superior visibility.

Minor critique 2:

Figure 9: this figure is a nice and valuable addition to illustrate the core principle of the paper. However, according to the presented data, the deviation from the “core” neutrotime programme as a consequence of inflammation does mainly occur at the mature end of the neutrotime scheme (high neutrotime-S score). This should be better illustrated in the figure. As it stands now, it seems that deviations into the direction of inflammatory tissue neutrophils equally occur at all stages of neutrotime. As far as I understood the paper, this is not the case and not supported by the data.

RESPONSE: As noted above, we do indeed intend the sense that neutrophils of distinct phenotypes could potentially be elicited anywhere along the neutrotime continuum, as supported by the more immature phenotype of blood neutrophils with inflammation. We also clarified the legend, as noted above.

With regards

Sven Brandau

Reviewer #4:

In their revised version, the authors added new analysis (i.e. human neutrophils analysis, mRNA-protein comparison) and clarified the overall message of the paper. However, my previous criticism remains unanswered. I'm very concerned about the novelty of the manuscript and the support for the main claim: 'It is this rejection of the subset model that represents the most novel and important conclusion of our study'. I am afraid that the authors do not provide ample evidence to support their claim. Moreover, the authors claim in the letter to the reviewers that 'While we expect that cells at different stages of the neutrotime continuum will have different functions, we make no specific claims to that effect here that might benefit from the functional experiments described'. It is my opinion that the lack of any functional experiments to support their claim leaves this study very descriptive and do not add useful information to the field.

RESPONSE: We appreciate that the Reviewer noted that we have “added new analysis (i.e. human neutrophils analysis, mRNA-protein comparison) and clarified the overall message of the paper”, although we regret that not all of our responses were considered as positively. We recognize that, as scientists engaged together in an effort to understand health and disease, we can expect that there are areas in which it is reasonable to disagree. We are however encouraged that the other three Reviewers did not share Reviewer 4's overall concerns.

With respect to subsets vs. continuum: We show that the data can indeed be induced to subset, but that the number of subsets varies widely within the range of reasonable clustering assumptions. We conclude that the division of neutrophils into discrete “types” would therefore require an unacceptably subjective choice among clustering assumptions. As an alternative, we show that a continuous model accommodates all the findings, without branching that would suggest distinct sub-lineages. Further, using RNA velocity, we show that the continuum is temporally ordered. We propose therefore that neutrophils are a single lineage, within which differences at the level of gene expression reflect maturational stage rather than distinct subsets.

We agree with the Reviewer that more work is required to describe how neutrophils at different stages of neutrotime differ in function. This project will require the development of surface markers characteristic of each transcriptional phase – a challenging task given the marked variation between transcript/protein that we show in Extended Data Figure 11 – followed by sorting and functional studies under experimental conditions *in vitro*

and *in vivo*. We are enthusiastic about these studies but recognize that they will take years to do well. In the meantime, we submit that our demonstration that neutrophils represent a single continuum represents a substantial advance.

A non-comprehensive list of comments is below.

1. The fact that the authors apply Monocle and diffusion maps algorithms to their data, that display a continuum is not surprising. This is the heart of these algorithms. Many other datasets can be depicted similarly, but the real challenge is to show that indeed this data representation can be validated. Otherwise, strong claims made by the authors remain ambiguous and, unfortunately, not concise.

RESPONSE: As described above, we were driven to Monocle and diffusion maps by the evident failure of clustering to achieve a stable result. While these approaches do indeed seek continuities in datasets, both methods are also designed to detect branching within continuities, which was not observed in our dataset. For examples of branching detected by monocle or diffusion maps see the following non-exhaustive list of studies:

Athanasiadis, E.I., J.G. Botthof, H. Andres, L. Ferreira, Pietro Lio, and A. Cvejic. 2017. Single-cell RNA-sequencing uncovers transcriptional states and fate decisions in haematopoiesis. *Nat Comms.* 8:1–11. doi:10.1038/s41467-017-02305-6. *Figures 1b, 2a, 3c (Monocle) – Zebrafish hematopoiesis with initial branching from HSPCs to either erythroid/megakaryocyte or myeloid cells, then branching of these lineages to erythrocytes and megakaryocytes or monocytes and neutrophils, respectively.*

Haber, A.L., M. Biton, N. Rogel, R.H. Herbst, K. Shekhar, C. Smillie, G. Burgin, T.M. Delorey, M.R. Howitt, Y. Katz, I. Tirosh, S. Beyaz, D. Dionne, M. Zhang, R. Raychowdhury, W.S. Garrett, O. Rozenblatt-Rosen, H.N. Shi, O. Yilmaz, R.J. Xavier, and A. Regev. 2017. A single-cell survey of the small intestinal epithelium. *Nature.* 551:333–339. doi:10.1038/nature24489. *In particular Extended Data Figure 4b describing proximal vs. distal commitment (diffusion maps).*

Hamey, F.K., S. Nestorowa, S.J. Kinston, D.G. Kent, N.K. Wilson, and B. Göttgens. 2017. Reconstructing blood stem cell regulatory network models from single-cell molecular profiles. *Proc. Natl. Acad. Sci. U.S.A.* 114:5822–5829. doi:10.1073/pnas.1610609114. *In particular Figures 1b and 2 showing a megakaryocyte-erythroid progenitor (MEP) vs. lymphoid-primed multipotent progenitor (LMPP) bifurcation from hematopoietic stem cells (HSC) using diffusion maps.*

Haghverdi, L., M. Büttner, F.A. Wolf, F. Buettner, and F.J. Theis. 2016. Diffusion pseudotime robustly reconstructs lineage branching. *Nat Meth.* 13:845–848. doi:10.1038/nmeth.3971. *In the original diffusion map publication, the algorithm was used to describe the endothelial vs. erythroid branches in early hematopoiesis (Fig. 1).*

We highlight that Monocle fits the principal graph on a per-partition basis. Our unsupervised approach led to the generation of a single trajectory, again providing no evidence of informative subclusters. Importantly, we achieved a very similar result by projecting the cells using only principal component analysis, followed by projecting a principal curve through the data points (Extended Data Fig. 5). Finally, we could obtain strong support for a continuum by RNA velocity analysis, an independent method that yielded the same result. Thus, while we agree with the Reviewer that it is important to be alert for results that reflect simply the methods chosen, we believe that no such concern is sustainable with respect to neutrotime.

2. Furthermore, in light of recent publications, especially- <https://doi.org/10.1038/s41590-020-0736-z> (cited in the manuscript), rejection of the subset model should be better discussed. In the mentioned paper, the exact tissues were studied in mice, revealing a trajectory of subtypes of neutrophils. Their model was then validated in the context of infection and transcription factors, further questioning the novelty of the current manuscript.

RESPONSE: As the Reviewer notes, we have cited the referenced manuscript, which was evidently under review contemporaneously with ours. We consider the two manuscripts to provide complementary datasets

and analyses. While the data from healthy tissues are largely concordant, our conclusions are distinct because our analytical approaches differ. Specifically, these authors retained the clustering approach, an analytical commitment which forced them to postulate a large multiplicity of neutrophil subgroups. By rejecting this commitment and proceeding to a continuum model, we find that this complexity resolves into a single unitary spectrum. We favor our approach, which we believe will serve as a model for similar analyses going forward; however, we recognize also that multiple analytical strategies are internally coherent. Further, the studied perturbations do not overlap (*E. coli* bacteremia in the referenced study vs. IL-1 β peritonitis, IL-1 β pneumonitis, and K/BxN serum transfer arthritis in ours), so that the two studies together provide an informative overview of neutrophil transcriptomes at steady state, as well as in septic and sterile inflammation. We hope that the two papers will be read together and anticipate that fruitful discussion will ensue.

3. The authors present an alternative model here, the continues model for neutrophils. However, the results themselves seems not to support this model:

a. The claim for 'instability of the clusters' cannot be deduced from the results in supp fig. 2a. Obviously, changes of the parameters will affect the number of clusters. However, P1-3 seems very distinct.

RESPONSE: As the Reviewer notes, P1-P3 are distinct from each other, as would be any set of clusters derived mathematically from the dataset on the basis of differentially expressed genes. The problem, in our view, is that equally acceptable assumptions yield sets of anywhere between 3 and 23 clusters, in which each cluster is also distinct from every other. Choosing among these organizational options is therefore simply subjective, which we found unsatisfactory. By considering the possibility that the dataset is continuous, we find that we are able to account for the variability without clusters. This does not mean that groups cannot be clustered for convenience, but does suggest that it is likely a mistake to "reify" one or another clustering scheme as though it identified discrete types of neutrophils in the way that CD4+ T cells are a type of T cell.

b. It will be hard to convince the TF data in Figure 6 follows a continuum rather than subsets. Same claim to the IFN-I data. If the authors want to claim continuum, they need to provide evidence.

RESPONSE: We suspect that there is a misunderstanding here. Our argument in favor of a continuum is completed by this stage in the manuscript. In Figure 6, we simply show that transcription factors change progressively along the neutrotime continuum, and that no single TF seems to drive the spectrum.

c. The validation of the neutrotime model (fig. 5) does not validate a continues pattern. First, the correlation between neutrotime and neutrotime-s genes seems problematic. Despite the high Spearman correlation coefficient, there are ranges that are dramatically modified. For example, the whole range of 0.25-0.75 in neutrotime score will get a very small score range in neutrotime-s of ± 0.75 . Furthermore, the bulk studies do support different expression of the neutrotime-s genes, however the validation should be to continues expression, not to the expression of the selected genes.

RESPONSE: As the Reviewer will appreciate, we could not extract a quantitative neutrotime score directly from the UMAP algorithm, and so needed to derive a sub-score (neutrotime-S) in order to apply the neutrotime scale to other datasets. Our expectation was never that the correlation should be perfect, but only that it would be good enough to enable an apples-to-apples comparison between datasets. For this purpose, it is sufficient that genes exhibit comparable rank order. Fortunately, we were successful in this effort, achieving a Spearman of 0.905 (Figure 5b). As confirmation of the success of this approach, we show in Figure 5c that our score fits extremely well with published data from Evrard et al. ($P < 0.0001$).

d. In figure 2G, the authors define a bone marrow to blood transition: "Blood neutrophils in healthy mice overwhelmingly occupy the most mature end of the spectrum. A sharp increase in neutrotime accompanies the transition from marrow to blood, suggesting that this transit event or other factors associated with circulation rapidly induce final neutrophil maturation." What is the support for this with a continuum as opposed to clustered data?

RESPONSE: Thank you for this question. We show in Figure 2g that there is a distinct increment in neutrotime between the most mature neutrophils in the marrow and the population of neutrophils circulating in blood.

e. In their infection model, the authors use the neutrotime genes. Where is the advantage of a continuum analysis here? Genes from P1-P4 seem to have been just as relevant.

RESPONSE: We derived neutrotime using healthy marrow, blood, and spleen. Applying the model to the inflammation models, we are able to show that the cells in inflamed tissue represent the most mature end of the neutrophil spectrum but also how the cells change upon arrival, depending upon context. The genes from P1-P4 are of course the same genes, but the neutrotime model no longer forces us to ask how a P1 cell might react compared to P2 cell and so on, since in fact they represent a continuous spectrum.

We thank the Reviewers and the Editors for their detailed attention to our manuscript and hope that the work will now be found suitable for publication, enabling a continuation of this discussion in the public space.

Sincerely yours,

Peter A. Nigrovic, MD for the authors

REVIEWERS' COMMENTS

Reviewer #3 (Remarks to the Author):

I thank the authors for further improving this excellent piece of work. All my comments are now addressed in full.

I am optimistic that the manuscript, if/when published, will stimulate fruitful discussions in the field.

Reviewer #4 (Remarks to the Author):

At this point I am afraid that the points we raised were NOT addressed satisfactorily.

My major concern is the following: there is no problem with publishing similar works at the same time, this happens in science and broadens our understanding of the scientific question. However, if the main claim of the same question is different between current and already published works, here- rejecting the clusters model for neutrophils subset and describing neutrophils as continuum model, the new claim should be well justified. Otherwise, I do not see the motivation for publishing results that contradict the current knowledge and are neither convincing from analysis aspects nor supported by experimental tools.

The heart of the problem is that the genes that determine clusters P1-P4 (as appears in fig. 1c) overlap with the same genes that determine neutrophil-S (genes from fig 3b). How can this reject the clusters model?

Of note- in the authors response: "We show that the data can indeed be induced to subset, but that the number of subsets varies widely within the range of reasonable clustering assumptions. We conclude that the division of neutrophils into discrete "types" would therefore require an unacceptably subjective choice among clustering assumptions.". ALL clustering analysis require subjective choice. Indeed - in some cases it is not possible to make reasonable assumptions - but I am afraid the authors have not convinced me this is the case here. Indeed - a recent paper shows there is, and also validates the results.